# Effects of warming and nitrogen deposition on species and functional diversity of plant communities in the alpine meadow of Qinghai-Tibet Plateau

**Xuemei Xiang, Kejia De** [ID]**\*, Weishan Lin, Tingxu Feng, Fei Li, Xijie Wei**

College of Animal Husbandry and Veterinary Science, Qinghai University, Xining, Qinghai Province, China

\* dekejia1002@163.com

## Abstract

Plant species and functional diversity play an important role in the stability and sustainability of grassland ecosystems. However, the changes and mechanisms of plant species and functional diversity under warming and nitrogen deposition are still unclear. In this study, we investigated the plant and soil characteristics of alpine meadows on the Qinghai-Tibet Plateau to explore the changes in species and functional diversity of plant communities under warming and nitrogen deposition, as well as their interrelationships and key determinants. The results showed that warming, nitrogen deposition, and their interactions had significant effects on plant species diversity (plant Shannon-Wiener index) and functional diversity (functional richness index, functional differentiation index, functional dispersion, and Rao's quadratic entropy index). With the increase of warming and nitrogen deposition, the Shannon-Wiener index of plants increased first and then decreased. The plant functional richness index, functional diversity index, functional dispersion index, and Rao's quadratic entropy index showed a decreasing trend. At the same time, with the increase in temperature and nitrogen deposition, the relationship between plant species diversity index and functional diversity index in the alpine meadow of Qinghai-Tibet Plateau gradually weakened. Redundancy analysis and structural equation modeling showed that both warming and nitrogen deposition had significant negative effects on the plant species diversity index and plant functional diversity index. Plant factors (Grasses importance value, leaf nitrogen weighted mean, specific leaf area-weighted mean, leaf area-weighted mean, and leaf weight weighted mean) and soil environmental factors (soil total nitrogen and soil carbon-nitrogen ratio) directly or indirectly affect plant community diversity under warming and nitrogen deposition.

## Introduction

Global surface temperatures are projected to rise by approximately 4°C by 2100 [1]. Simultaneously, anthropogenic nitrogen emissions and deposition are increasing rapidly worldwide [2]. In the past few decades, the combined effects of climate change factors such as

**Data availability statement:** All relevant data are within the manuscript and its Supporting Information files.

**Funding:** This research is supported by the key R & D and transformation plan project of Qinghai Provincial Department. of Science and Technology (2024-NK-137) (awarded to Kejia De); research and demonstration of one-year-old forage seed reproduction and silage processing and storage technology in Chengduo County (2024-NK-P28) (awarded to Xijie Wei); 2025-2026 China Association for Science and Technology Young Talent Promotion Project doctoral program (awarded to Xuemei Xiang); and funded by the Ministry of Education Field Scientific Observation and Research Station of Sanjiangyuan Ecosystem (awarded to Kejia De).

**Competing interests:** The authors have declared that no competing interests exist.

temperature rise and nitrogen deposition have led to the degradation of the function and structure of grassland ecosystems, resulting in accelerated desertification, loss of biodiversity, and changes in carbon balance [3]. The loss of biodiversity not only means species extinction but also threatens food security, social stability, and human survival through the food chain and web.

Plant community diversity is an important part of biodiversity [4]. Plant species diversity reflects the complex relationship between biology, environment, and biological resource richness [5]. Research has shown that global warming significantly reduces plant species richness and diversity in terrestrial ecosystems [6]. Different plant functional groups respond differently to climate warming [7]. The effects of nitrogen deposition on plant species diversity vary depending on the nitrogen use efficiency and adaptability of the plant community [2,8,9]. In the study of temperate grassland and semi-arid grassland, it was found that plant species richness showed a downward trend under nitrogen deposition [10,11]. With the increase in temperature and nitrogen deposition, the number of vascular plants is generally increasing, which may be because temperature and nutrients are the main limiting factors of alpine ecosystems [12]. Plant functional diversity takes into account the redundant and complementary functions within the plant community, as well as the response of functional traits to environmental stress or disturbance [13]. Species diversity and functional diversity are key determinants of grassland ecosystem stability [14,15]. Therefore, exploring the feedback mechanism of plant community diversity on climate warming and nitrogen deposition is of great scientific value and practical guiding significance for scientifically understanding the maintenance mechanism of grassland ecosystem diversity on the Qinghai-Tibet Plateau under climate change and guiding grassland management.

The relationship between plant community species and functional diversity helps to explore the response of ecosystems to environmental changes [16], reveal the mechanism of species coexistence in plant communities, and determine ecosystem management policies and rational use of grassland resources [17]. Species diversity can increase the functional diversity of vegetation communities [15]. Functional diversity is mainly determined by interspecific interactions and niche complementarity, affecting ecosystem functions [18]. The response of plant community species diversity to climate change primarily depends on its biological characteristics [19]. These characteristics are mainly determined by the physiological traits and competitive abilities of the dominant species [20,21]. Studies have shown that nitrogen enrichment enhances specific leaf area, and leaf nitrogen concentration in plant species [22,23]. However, nitrogen enrichment reduces the abundance of plants with resource-conservative strategies [24]. Beyond leaf economic traits, functional traits such as plant height also influence the response to nitrogen enrichment. Nitrogen enrichment increases aboveground biomass and canopy cover, shifting competition among plants from nutrient-based to light-based [25]. Taller plants capture more light resources, leading to asymmetrical competition that can alter species diversity in plant communities [26]. In alpine meadow ecosystems, there is a linear relationship between plant functional diversity and species diversity [27]. In summary, due to the influence of species composition, environmental resources, and land use, the relationship between species diversity and functional diversity has not yet reached a consensus [13].

In addition, soil physicochemical properties also affect plant species distribution and diversity [28]. Soil factors determine which species or functional traits are retained, and this variation in plant adaptation to the environment influences inter-community competition, thus affecting plant community structure [29]. Studies have shown that as soil moisture and nutrient availability increase, species with lower resource acquisition abilities but higher resilience tend to dominate [30]. In temperate grassland, soil pH, total nitrogen, and nitrogen-phosphorus ratio are the key factors to explain plant functional diversity [31]. Studies have

also shown that soil nitrogen content in the short term and soil phosphorus content in the long term affect plant species richness in the long term [32]. Studies on the Qinghai-Tibet Plateau have shown that soil organic matter, available nitrogen, and phosphorus significantly affect plant diversity in alpine meadows [33]. In the semi-arid region of the Qinghai-Tibet Plateau, the depth of permafrost active layer and soil water content are important factors affecting the distribution of plant communities [34]. Combined with the above, the key soil physical and chemical factors affecting the diversity of plant communities in different research areas are different. Therefore, it is necessary to clarify the relationship between plant community diversity and soil physical and chemical properties in alpine meadows on the Qinghai-Tibet Plateau, which will help to understand the overall function of grassland ecosystems better.

The Qinghai-Tibet Plateau is a significant center for the origin and differentiation of global species and is one of the most biodiversity-rich regions in alpine areas. However, climate change-induced biodiversity alterations threaten the ecological security barrier function of the Qinghai-Tibet Plateau, which in turn affects ecological security and sustainable development in China, Asia, and the Northern Hemisphere, posing challenges to biodiversity conservation [35,36]. Therefore, the Qinghai-Tibet Plateau is an ideal area to explore climate change's impact on alpine regions' biodiversity [37]. Studies have shown that climate warming and nitrogen deposition can have substantial impacts on the structure and function of alpine meadow ecosystems, such as reducing biodiversity, altering carbon cycles, and changing grassland multifunctionality [38,39]. The increase in temperature can affect soil moisture and then change soil available nutrients, thereby affecting nitrogen input. Therefore, multiple climatic factors can lead to complex interaction effects, and these effects cannot be predicted by the sum of the effects of two single factors [40]. Therefore, this study takes the alpine meadow on the Qinghai-Tibet Plateau as the research object to explore the response mechanism of species diversity and functional diversity of alpine meadow to climate warming and nitrogen deposition. The research objectives are as follows: (1) To study the changes in species diversity and functional diversity in alpine meadows under warming and nitrogen deposition;(2) to Determine whether there is a relationship between species diversity and functional diversity under warming and nitrogen deposition;(3) Explore the key factors affecting species diversity and functional diversity. The results of this study have clarified the response mechanism of plant biodiversity to climate warming and increased nitrogen deposition, which is of great significance for understanding the biodiversity protection of the Qinghai-Tibet Plateau and even the global alpine ecosystem and the maintenance and improvement of the ecological security barrier function.

## Materials and methods

### Study area overview

The study was conducted at the Chengduo substation of the Sanjiangyuan Grassland Ecosystem National Field Scientific Observation and Research Station in Qinghai, China (33° 24′30″ N, 97° 18′00″ E), at an elevation of 4,270 meters. The region experiences a typical plateau continental climate, with an annual average temperature ranging from -5.6°C to 3.8°C and an average annual precipitation of 562.2 mm, 75% of which occurs between July and September. The dominant plant species include *Kobresia humilis* (C.A.Mey ex Trauvt) Serg. and *Kobresia pygmaea* Clarke from the Cyperaceae family, as well as *Elymus nutans* Griseb. And *Poa annua* L. from the Gramineae family.

### Experimental design

The field experiment was initiated in May 2023 within a 50m × 50m fenced area. Studies have shown that the Tibetan Plateau is expected to face an additional warming of up to 2.0°C by

2035 and an additional warming of 4.9°C by 2100 [41]. To simulate different levels of warming and adjust the diameter of the top and bottom of OTC, four temperature treatments (no warming and three warming treatments) were selected: no warming (W0), warming 1 (W1, top diameter of 1.5m, bottom diameter of 1.95m), warming 2 (W2, top diameter of 1m, bottom diameter of 1.45m), warming 3 (W3, top diameter of 0.5m, bottom diameter of 0.95m). The height of the open-top growth chamber (OTC) was 40 cm. Soil and air temperatures are continuously measured from June to October using the HOBOS-TMB-M006 temperature sensor (Temperature Smart Sensor, HOBO, USA). The warming effect is shown in Table 1.

The environmental nitrogen deposition in the Tibetan Plateau is about 8 kg N ha$^{-1}$ yr$^{-1}$, mainly in the form of $NH_4$-N and $NO_3$-N [42]. Therefore, three nitrogen deposition treatments were selected to simulate future atmospheric N deposition: N0 (0 kg ha$^{-1}$ yr$^{-1}$), N16 (16 kg Nha$^{-1}$ yr$^{-1}$), and N32 (32 kg Nha$^{-1}$ yr$^{-1}$). Ammonium nitrate was used to simulate nitrogen deposition in the growing season. $NH_4NO_3$ was dissolved in water and evenly sprayed on the plot, and the same amount of water was also sprayed in the control. A randomized block design was used in this experiment. Four replicates were set in the plot, and 12 treatment levels were set in each plot, W0N0 (no warming and nitrogen addition), W0N16, W0N32, W1N0, W1N16, W1N32, W2N0, W2N16, W2N32, W3N0, W3N16, W3N32. A total of 48 samples were obtained.

## Vegetation survey

From 2023 to mid-August 2024, a sample plot survey was conducted at the peak of the vegetation growing season to obtain vegetation data. The sampling method was repeated four times, and the sampling area was 0.5 m × 0.5 m. According to the differences in life forms of different plant species in alpine communities on the Qinghai-Tibet Plateau, all species in the quadrat were divided into grass functional groups, sedge functional groups, and forb functional groups [43]. The survey included the number of plant species (species richness), plant height, plant coverage, and plant aboveground biomass in each quadrat. Plant coverage was measured by the percentage of the projected area of each species to the entire plot area. Plant height determination refers to the determination of the natural height of all species in the quadrat. Five plants were randomly selected from each species to determine the height. The aboveground biomass of the plant is cut from all the plants on the ground and the plant samples at 65 °C to constant weight.

## Determination of soil physical and chemical properties

In August 2023-2024, samples were collected during the plant growth season, and a 0.5 × 0.5 m sample box was randomly placed in each plot. Five soil cores with a diameter of 3 cm and a depth of 30 cm were drilled in each sample. After mixing them, the roots and stones of the plants were removed with a 2 mm mesh sieve to the determine soil's physical and chemical

**Table 1. Specifications and effects of heating devices.**

| Heating Gradient (°C) | Top Diameter (m) | Bottom Diameter (m) | Height (m) | Temperature(°C) | |
|---|---|---|---|---|---|
| | | | | Air temperature at 10 cm above the ground | Soil temperature at 5 cm below the ground surface |
| W1 | 1.5 | 1.94 | 0.4 | 0.47 | 0.61 |
| W2 | 1 | 1.45 | 0.4 | 0.92 | 1.09 |
| W3 | 0.5 | 0.95 | 0.4 | 1.44 | 1.95 |

Note: The warming effect of W1, W2, and W3 is compared with that of W0 (no warming). The data in the table is the mean for June to October.

properties. Soil water content (SWC) was measured by oven-drying method (105 °C, 24 h). Soil pH (pH) was measured by the potentiometric method (the soil-water ratio was 1: 2.5). Soil bulk density (SBD) was measured by the ring knife method. Soil total phosphorus (STP) was determined by the Mo-Sb colorimetric method. Soil ammonium nitrogen (SAN) was determined by the potassium chloride extraction-indophenol blue colorimetric method [44]; soil nitrate nitrogen (SNN) was determined by a dual-wavelength ultraviolet spectrophotometer [29].

## Data calculation

**Species diversity of plant community.** Species diversity within the community was evaluated using alpha diversity indices, including 1) Species richness index (R); 2) Shannon-Weiner index (H); and 3) Pielou's evenness index (J). The formulas are as follows [45]:

Importance value (IV): is the mean value of the sum of relative height (RH), relative coverage (RC), and relative biomass (RB).

$$IV = \frac{RH + RC + RB}{3}$$

Species richness Index:

$$R = S$$

Shannon-Weiner Index:

$$H = -\sum_{i=1}^{s} p_i^2$$

Pielou's evenness index: $J = H \,/\, lnS$

In these formulas, S represents the number of species, Pi denotes the proportion of individuals of species iii relative to the total number of individuals.

**Plant functional diversity.** From 2023 to August 2024, during a vegetation survey, six plant species were selected from the recorded 15 species for the measurement of functional traits. These species included Sedges (*Kobresia humilis*), Grasses (*Poa pratensis*), Forbs (*Potentilla anserina*, *Taraxacum lugubre*, *Gentiana futtereri*, and *Oxytropis ochrocephala*). The reason for determining these target species is that they are present in each plot, and these six plant species account for 80% of the total plant species coverage [46] (Fig 1).

Mature individuals with good growth were randomly selected from each treatment to measure functional traits. Five plants were selected from each plant species in each quadrat for plant height observation records. After observing the height of the plant, the leaves were cut off and scanned using a leaf area meter (Yaxin-1241 leaf area meter) to obtain leaf area, leaf length, and leaf width. Due to the small leaves of Kobresia pygmaea, to reduce the measurement error, 30 plants were selected to calculate the total area, and then the area of each leaf was calculated. After the measurement, a single plant leaf was placed in an oven, dried at 65 °C for 48 h, and weighed using an analytical balance to obtain leaf dry matter content(LDMC) [47]. The specific leaf area was calculated by leaf area and leaf dry weight [48]. The formula is as follows:

$$SLA\left(g\,/\,cm^2\right) = LDMC\left(g\right)\,/\,LA(cm)^2$$

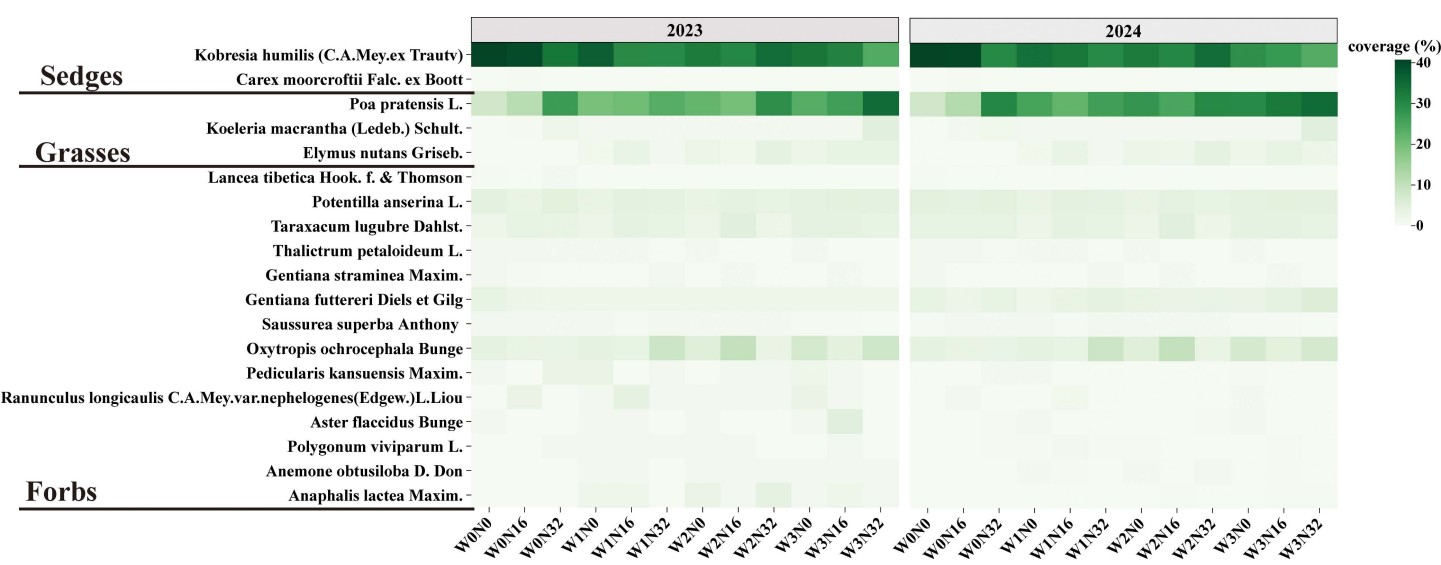

**Fig 1. Coverage of plant species in each plot.**

In this equation, SLA (g/cm²), LDMC (g), and LA (cm²) represent specific leaf area, leaf dry matter content, and total leaf area, respectively.

Plant leaf carbon and nitrogen content were determined using a German elemental analyzer (vario MACRO cube), while plant leaf phosphorus content was determined by boiling in $H_2SO_4$-$H_2O_2$ and measured using the molybdenum antimony anti-colorimetric method.

The community-weighted mean(CWM) traits for each subplot were quantified as follows:

$$Community \; weighted \; trait \; means = \sum_{i=1}^{S} Pi \times trait_i$$

In the equation, 's' represents the number of species in the plot, '$P_i$' denotes the relative abundance of species 'i' within the plot, and '$trait_i$' signifies the average trait value of species 'i' within that plot [47].

Based on the above, this study screened 9 plant functional traits (plant height, leaf area, leaf length, leaf width, specific leaf area, leaf dry weight, leaf total nitrogen, leaf organic carbon, leaf total phosphorus), and calculated plant functional diversity [49]. The five functional diversity indexes of functional richness (FRic), functional evenness (FEve), functional divergence (FDiv), functional dispersion (FDis), and Rao's quadratic entropy (RaoQ) were calculated.

## Statistical analysis

The data were evaluated for normality and mean square deviation before statistical analysis. The plant species diversity index, species richness index, species evenness index, the important value of Grasses, important value of Sedges, the important value of forbs, the weighted mean value of plant community functional traits, functional richness, functional evenness, functional divergence, functional dispersion, and Rao's quadratic entropy were used for descriptive statistics. SPSS 25.0 software was used for linear mixed model analysis, with warming, nitrogen deposition, and year as fixed factors and block group as random factors. To understand the effects of warming, nitrogen deposition, and year changes on species diversity, functional traits, and functional diversity of plant communities. IBM SPSS Statistics (SPSS

software version 25.0, Chicago, Illinois, USA) was used. Tukey's multiple comparison was performed, and the statistical test level was based on $P < 0.05$. The relationship between species diversity index and functional diversity index was fitted by Origin 2024. Species diversity and functional diversity were calculated using the ' vegan ' and ' FD ' packages in R version 4.2, respectively. Canoco 5.0 software was used to determine the effects of soil physical and chemical factors (soil water content, soil pH, soil bulk density, soil total nitrogen, soil total phosphorus, soil nitrate nitrogen, soil ammonium nitrogen, and soil organic carbon) on plant species diversity and functional diversity by redundancy analysis. Amos software (IBM SPSS Amos 24.0) was used to construct a structural equation model to explore the influence path of each factor on soil species diversity and functional diversity, and the fitting of the model was judged by combining the performance indicators of the model, including the ratio of chi-square to degree of freedom (less than 3 is good), P value (greater than 0.05 is good), comparative fitting index (CFI, greater than 0.9 is good), standardized root mean square residual (SRMR, less than 0.1 is acceptable) [50]. In the final model, any non-significant relationship between variables is deleted. Except for the chi-square test of model fitting in SEM was set to $P > 0.05$, the significance level of all statistical tests was set to $P < 0.05$. Except for structural equation modeling, all data analyses were performed in the R software environment (version 4.3.2). Before conducting structural equation model analysis, all response variables and predictors were standardized, and a z-score was used to represent the parameters to meet the hypothesis of the test used [51].

## Results analysis

### Effects of warming and nitrogen deposition on plant species diversity in alpine meadow

In 2023, compared with W0, the Shannon-Wiener diversity index increased significantly by 7.74% under W1 treatment, and decreased significantly by 2.61% under W3 treatment. The important value of grasses was significantly increased by 37.38% and 40.83% under W2 and W3 treatments, respectively. The importance value of forbs decreased significantly by 41.20% under the W3 treatment. The importance value of sedges decreased significantly by 24.68%, 37.74%, and 42.34% under W1, W2, and W3 treatments, respectively. Compared with N0, the importance value of grasses increased significantly by 18.52% and 69.72% under N16 and N32 treatments, respectively. The interaction of warming and nitrogen deposition will significantly affect the importance value of grasses, sedges, and forbs. The importance value of grasses reached the maximum under the W3N32 treatment, the importance value of sedges reached the minimum under the W3N32 treatment, and the importance value of forbs reached the minimum under the W2N32 treatment.

In 2024, Compared with W0, the Shannon-Wiener diversity index increased significantly by 1.76% and 3.53% under W1 and W2 treatments and decreased significantly by 3.02% under W3 treatment. The species richness index increased significantly by 27.5% under the W1 treatment. The important value of grasses was significantly increased by 55.42%, 111.78%, and 153.93% under W1, W2, and W3 treatments, respectively. The important value of forbs was significantly reduced by 34.29% and 43.18% under W2 and W3 treatments. The importance value of sedges decreased significantly by 19.54%, 33.37%, and 53.64% under W1, W2, and W3 treatments, respectively. Compared with N0, the Shannon-Wiener diversity index decreased significantly by 7.49% under the N32 treatment. The importance value of grasses increased significantly under N32 treatment. The importance value of sedges decreased significantly by 62.95% under N32 treatment. The interaction of warming and nitrogen deposition will significantly affect the Shannon-Wiener diversity index, the importance value

of grasses, and the importance value of sedges. The Shannon-Wiener diversity index and the importance value of forbs reached the minimum value under the W3N32 treatment, and the importance value of grasses reached the maximum value under the W3N32 treatment.

The results of the mixed linear model showed that interannual variation had no significant effect on plant species diversity index and plant functional groups(S1 Table, Fig 2).

## Effects of warming and nitrogen deposition on plant functional diversity in alpine meadow

In 2023, Compared with W0, the plant functional richness index was significantly reduced by 58.69% in the W3 treatment; the plant functional evenness index decreased significantly by 27.94%, 19.15%, and 9.91% under W1, W2, and W3 treatments. The plant functional dispersion index decreased significantly by 54.39%, 47.22%, and 32.64% under W1, W2 and W3 treatments. The Rao's quadratic entropy was significantly reduced by 57.62%, 67.83%, and 74.21% under W2 and W3 treatments. The plant height-weighted mean, leaf area-weighted mean, leaf length-weighted mean, leaf weight-weighted mean, and leaf nitrogen-weighted mean were significantly increased by 3.18%, 61.26%, 85.82%, 152.78%, and 13.64% under W3 treatment, respectively. The weighted mean of specific leaf area was significantly reduced by 54.83% under W3 treatment. Compared with N0, the plant functional dispersion index, and Rao's quadratic entropy were significantly reduced by 10.11% and 9.84% under the N32 treatment. The leaf area-weighted mean, leaf length-weighted mean, leaf weight-weighted mean, and leaf nitrogen-weighted mean were significantly increased by 125.99%, 131.14%, 81.29%, 62.52%, 97.72%, and 9.91% under N32 treatment, respectively. The interaction of warming and nitrogen deposition also significantly affected the plant functional richness index, plant functional dispersion index, Rao's quadratic entropy, leaf weight weighted mean, specific leaf area-weighted mean, and leaf nitrogen weighted mean. The plant functional richness index reached the minimum value under the W316 treatment, and the plant functional dispersion index and Rao's quadratic entropy reached the maximum value under the W3N32 treatment. The weighted mean of leaf weight and leaf nitrogen reached the minimum under the W3N32 treatment.

In 2024, Compared with W0, the plant functional richness index was significantly reduced by 44.73%, 51.13%, and 52.49% under W1, W2, and W3 treatments; the plant functional evenness index was significantly reduced by 14.01% and 16.16% under W2 and W3 treatments. The plant functional dispersion index was significantly reduced by 20.95%, 41.44%, and 43.55% under W1, W2, and W3 treatments. The quadratic entropy index of RaoQ significantly increased by 19.36%, 47.23%, and 51.22% under the W1, W2, and W3 treatments. The weighted mean of plant height increased significantly by 182.59% and 71.24% under W2 and W3 treatments. The weighted mean of leaf area increased significantly by 73.84% under W3 treatment. The weighted mean of leaf length increased significantly by 67.55%, 151.78%, and 230.31% in W1, W2, and W3 treatments, respectively. The weighted mean of leaf weight increased significantly by 144.27% and 209.28% under W2 and W3 treatments. The weighted mean of specific leaf area was significantly reduced by 13.38% and 2.99% under W2 and W3 treatments. The weighted mean of leaf nitrogen was significantly reduced by 10.36% and 16.66% under W2 and W3 treatments. The weighted mean of leaf carbon increased significantly by 26.67% and 28.03% under W1, W2 and W3 treatments. Compared with N0, the plant functional richness index was significantly reduced by 57.29% under N32 treatment. The weighted mean of plant height, weighted mean of leaf area, weighted mean of leaf length, weighted mean of leaf weight, weighted mean of specific leaf area, weighted mean of leaf nitrogen, and weighted mean of leaf carbon increased significantly by 63.53%, 13.32%, 93.65%, 159.04%, 20.76%, 4.71% and 26.07% respectively under N32 treatment. The plant functional

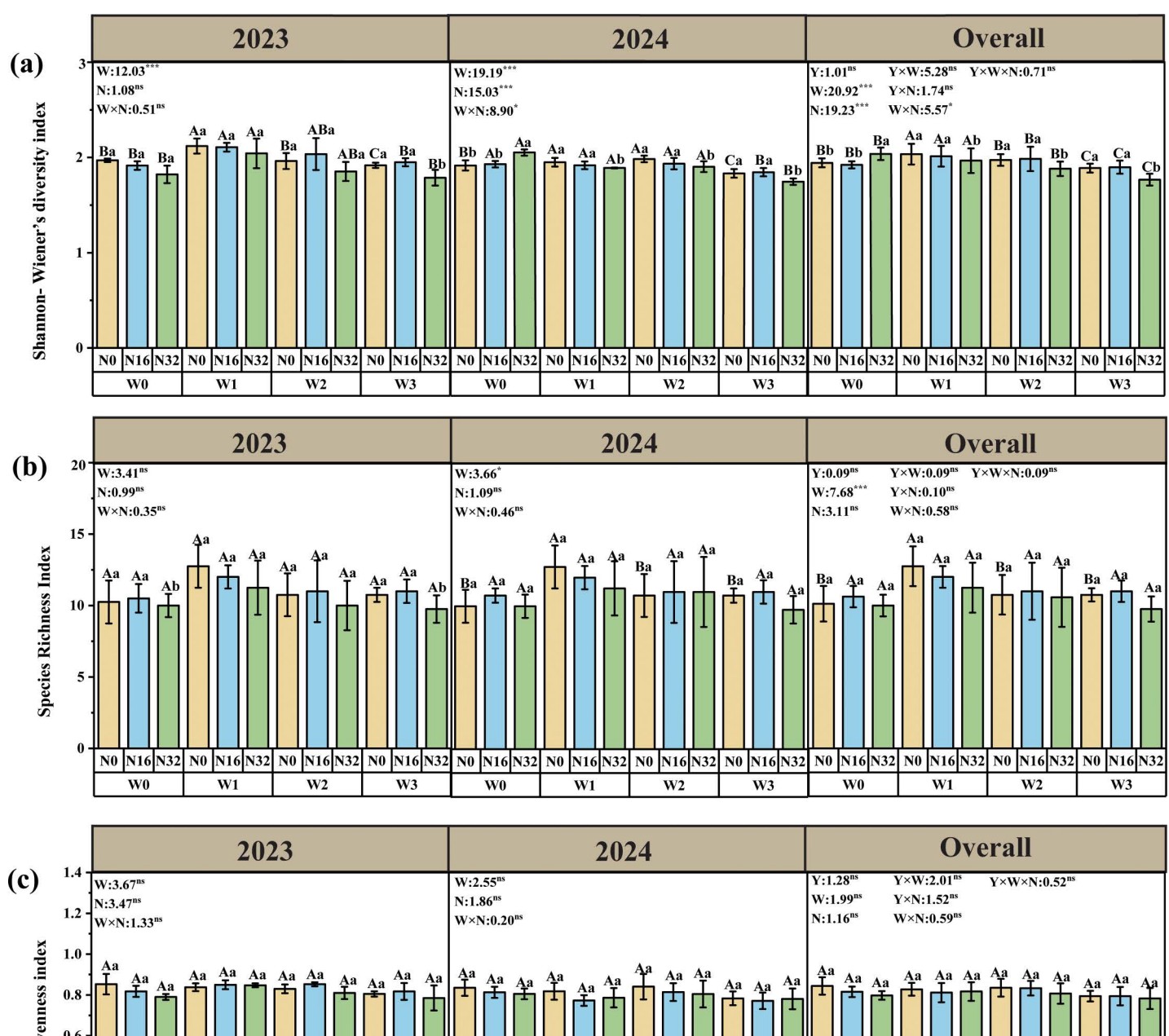

**Fig 2. Effects of warming and nitrogen deposition on plant species diversity.** Different lowercase letters indicated that there were significant differences in different nitrogen deposition treatments under the same warming level, and different uppercase letters indicated that there were significant differences in different warming treatments under the same nitrogen deposition level (ns, $P > 0.05$; *, $P < 0.05$; **, $P < 0.01$; ***, $P < 0.001$). The same below.

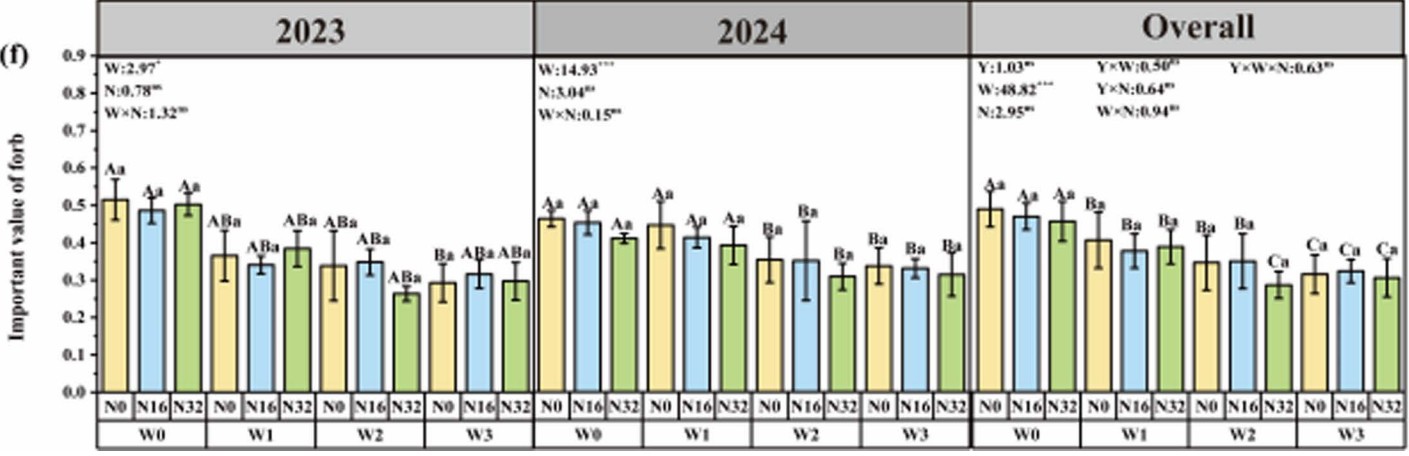

**Fig 2.** Continued.

dispersion index and Rao's quadratic entropy decreased by 10.27% and 11.45% respectively under N32 treatment. Under the W3N32 treatment, the plant function richness index, plant function dispersion index, and Rao's quadratic entropy reached the minimum value. Under the W2N32 treatment, the plant function uniformity reached the minimum value. The weighted mean of specific leaf area and leaf carbon reached the minimum value under W2N16

treatment. The plant height-weighted mean, leaf area-weighted mean, leaf length-weighted mean, leaf weight-weighted mean, and leaf nitrogen-weighted mean reached the maximum under W3N32 treatment. The results of the mixed linear model showed that interannual variation had no significant effect on plant functional diversity and mean functional traits (S2 and S3 Tables, Fig 3).

## The relationship between species diversity and functional diversity of plant communities under warming and nitrogen deposition

The fitting relationship and correlation coefficient between plant species and functional diversity index in alpine meadows were different under different warming and nitrogen deposition treatments, and the optimal fitting function was quadratic polynomial. There was a strong correlation between the plant functional diversity index and the Shannon-Wiener diversity index under W0N0, W0N16, W1N0, W1N16, W2N0, and W2N16 treatments. Under the treatments of W0N0, W0N16, W1N0, and W1N16, the plant functional dispersion index, Rao's quadratic entropy and Shannon-Wiener diversity index have a strong correlation. Under the treatments of W0N0, W0N16, W0N32, W1N0, W1N16, W1N32, and W2N0, the plant functional differentiation index, plant functional dispersion index, Rao's quadratic entropy, and species richness index have a strong correlation. Combined with the above results, with the increase of warming gradient and nitrogen deposition, the correlation coefficient between the plant Shannon-Wiener index and plant functional richness index, plant functional dispersion index, and Rao's quadratic entropy decreased gradually. At the same time, with the increase of the warming gradient, the correlation between the plant functional differentiation index, plant functional dispersion index, Rao's quadratic entropy, and species richness index gradually weakened (Fig 4).

## Response mechanism of plant species diversity and functional diversity to warming and nitrogen deposition

The relationship between plant species diversity index, functional diversity index, plant community functional traits, and soil physical and chemical properties was analyzed by redundancy analysis. Monte Carlo results showed that STN, CWM _ LNC, grasses, SC/N, SAN, and CWM _ SLA were the key factors in determining the plant species diversity index. Among them, the Shannon-Wiener diversity index and species evenness index of plants were negatively correlated with STN, SAN, CWM _ LNC, and CWM _ SLA, and positively correlated with Grasses and SC/N (Fig 5a, Table 2). The plant functional diversity index was mainly affected by grass, SC/N, CWM _ LDMC, SAN, CWM _ LA, SWC, SOC, SC/P, and SN/P (Fig 5b, Table 2).

The direct and indirect effects of plant and soil characteristics on plant species diversity and plant functional diversity under warming and nitrogen deposition were quantified by structural equation modeling (Fig 6). The results showed that warming and nitrogen deposition had significant negative effects on plant species diversity (path coefficients were -0.73 and -0.84, respectively). Warming and nitrogen deposition had a significant positive correlation with soil physical and chemical properties (path coefficients were 0.76 and 0.21, respectively), had a significant positive correlation with plant traits (path coefficients were 0.46 and 0.64, respectively), and negatively affected plant species diversity index through plant and soil characteristics (path coefficients were-0.39 and-0.26, respectively). Warming and nitrogen deposition had significant negative effects on plant functional diversity (path coefficients were-0.40 and-0.33, respectively), and negatively affected plant functional diversity index through plant and soil characteristics (path coefficients were 0.69 and 0.25, respectively). The

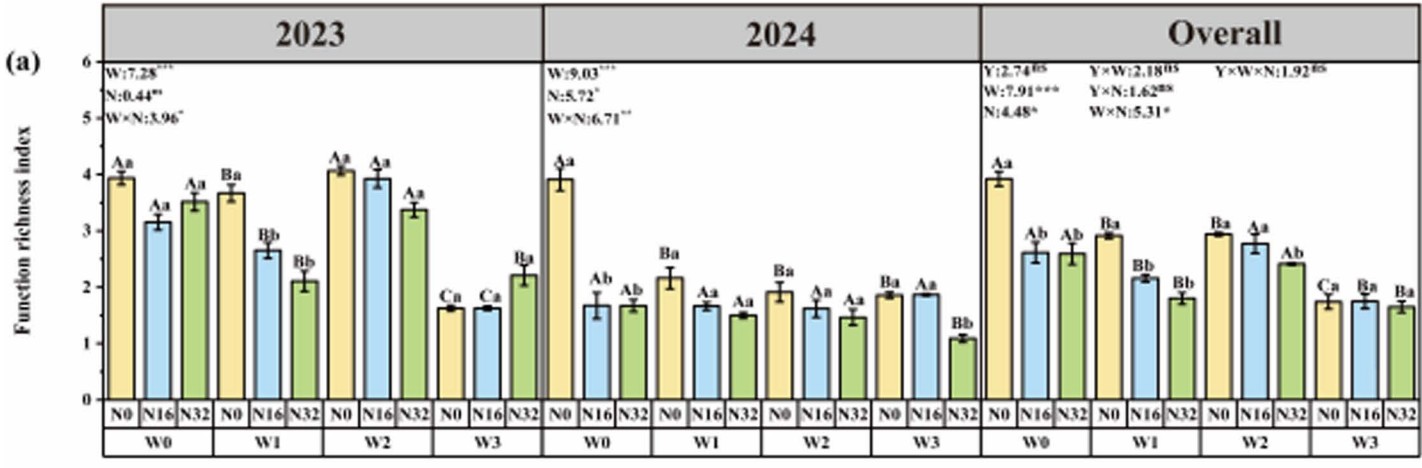

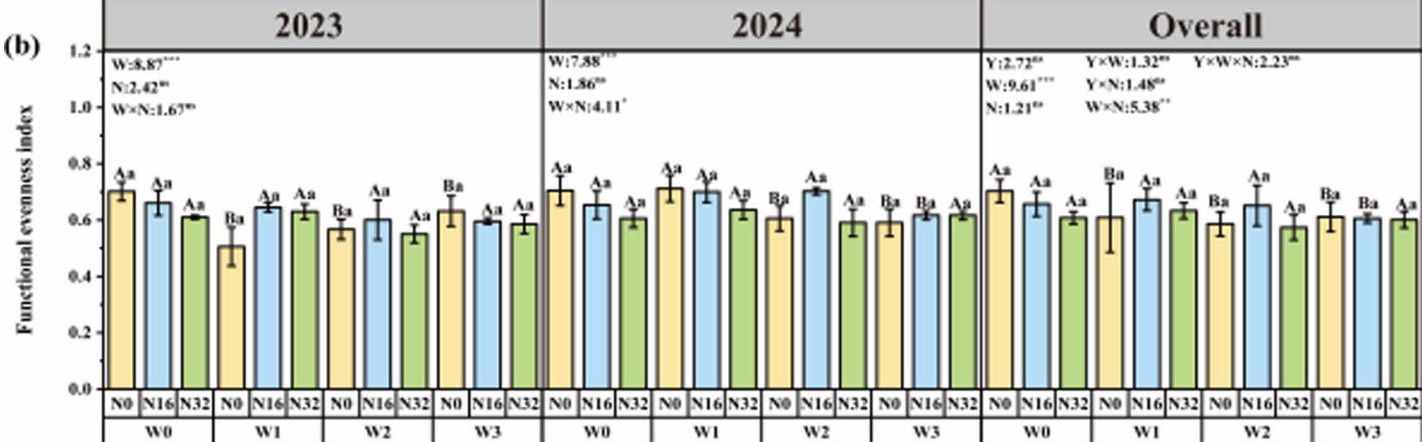

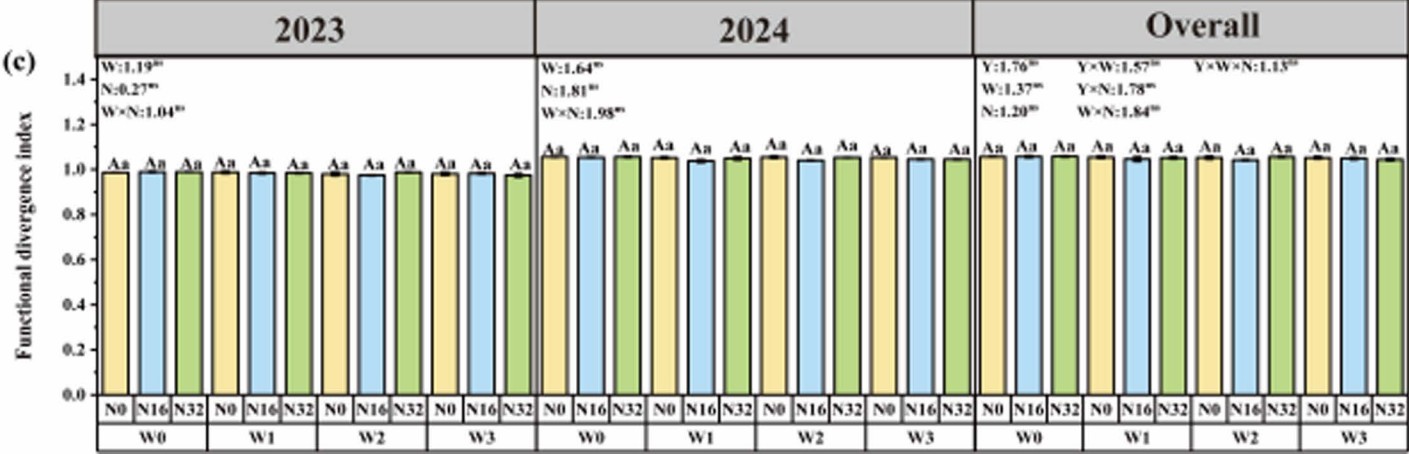

**Fig 3. Effects of temperature increase and nitrogen deposition on plant functional diversity.**

results also showed that plant functional diversity in alpine meadows was positively affected by plant species diversity under warming and nitrogen deposition (path coefficient was 0.56). In general, with the increase of warming and nitrogen deposition, the plant species diversity index and plant functional diversity index showed a decreasing trend. Soil factors (soil total

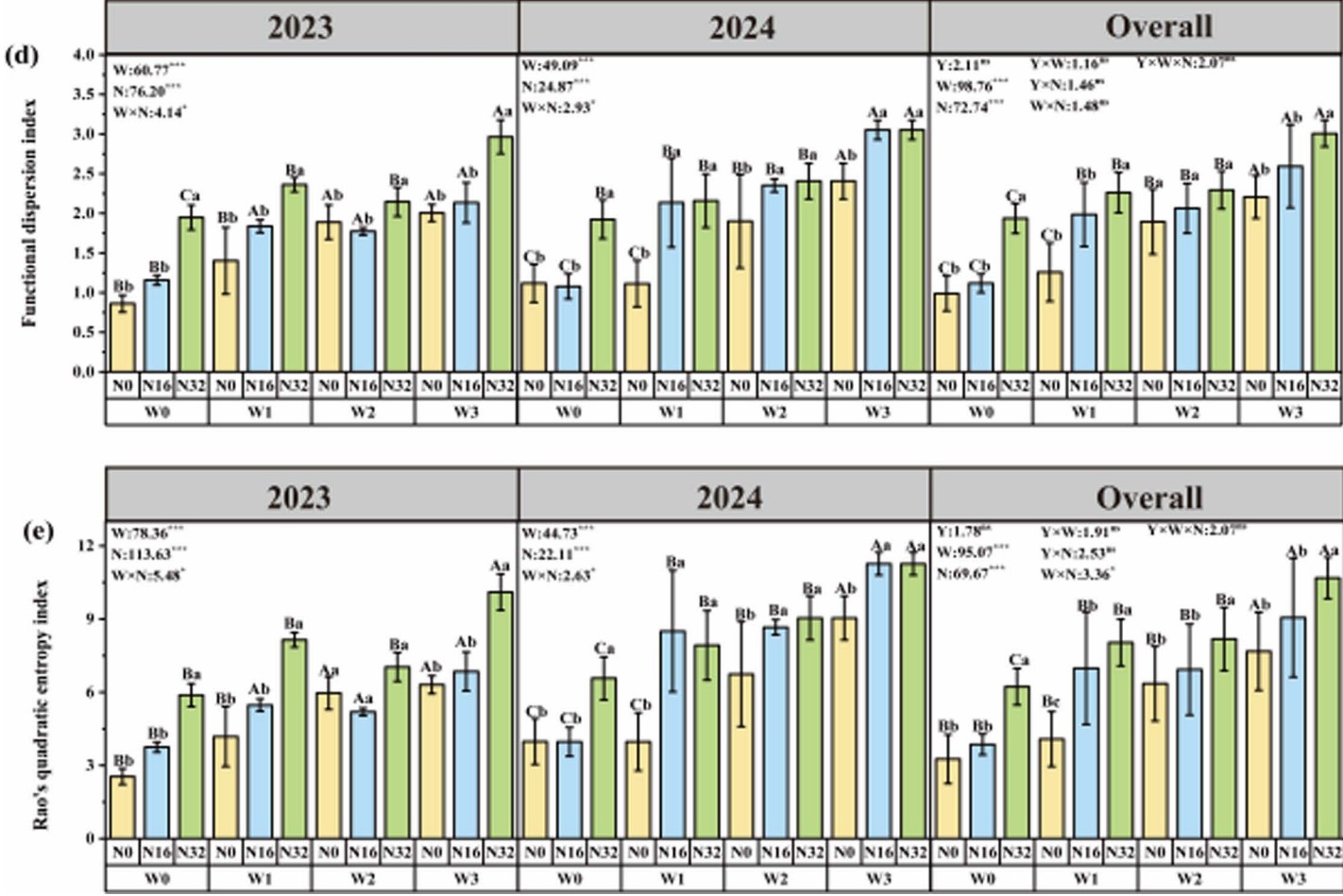

**Fig 3.** Continued.

nitrogen and soil carbon-nitrogen ratio) and plant factors (importance value of grasses, leaf nitrogen weighted mean, specific leaf area-weighted mean, leaf area-weighted mean, and leaf weight weighted mean) will have direct and indirect effects on plant species diversity and plant functional diversity through warming and nitrogen deposition.

## Discussion

### Effects of warming and nitrogen deposition on plant community species diversity

Plant species diversity is an important factor affecting the complexity and stability of community function [52]. The results of this study showed that the Shannon-Weiner index increased significantly and then decreased with the increase of the warming gradient, and the plant species richness index increased significantly under the W1 treatment. Studies have shown that warming has a significant negative impact on global grassland species richness but has no effect on plant Shannon-Wiener index and species evenness [53]. In the 4-year warming experiment on the alpine meadow, it was found that warming had no significant effect on species richness, Shannon diversity, and Shannon evenness [54]. Studies in the Loess Plateau have shown that low-level warming increases plant community species diversity, but high-level warming reduces plant species diversity [55]. This shows that compared with low-level

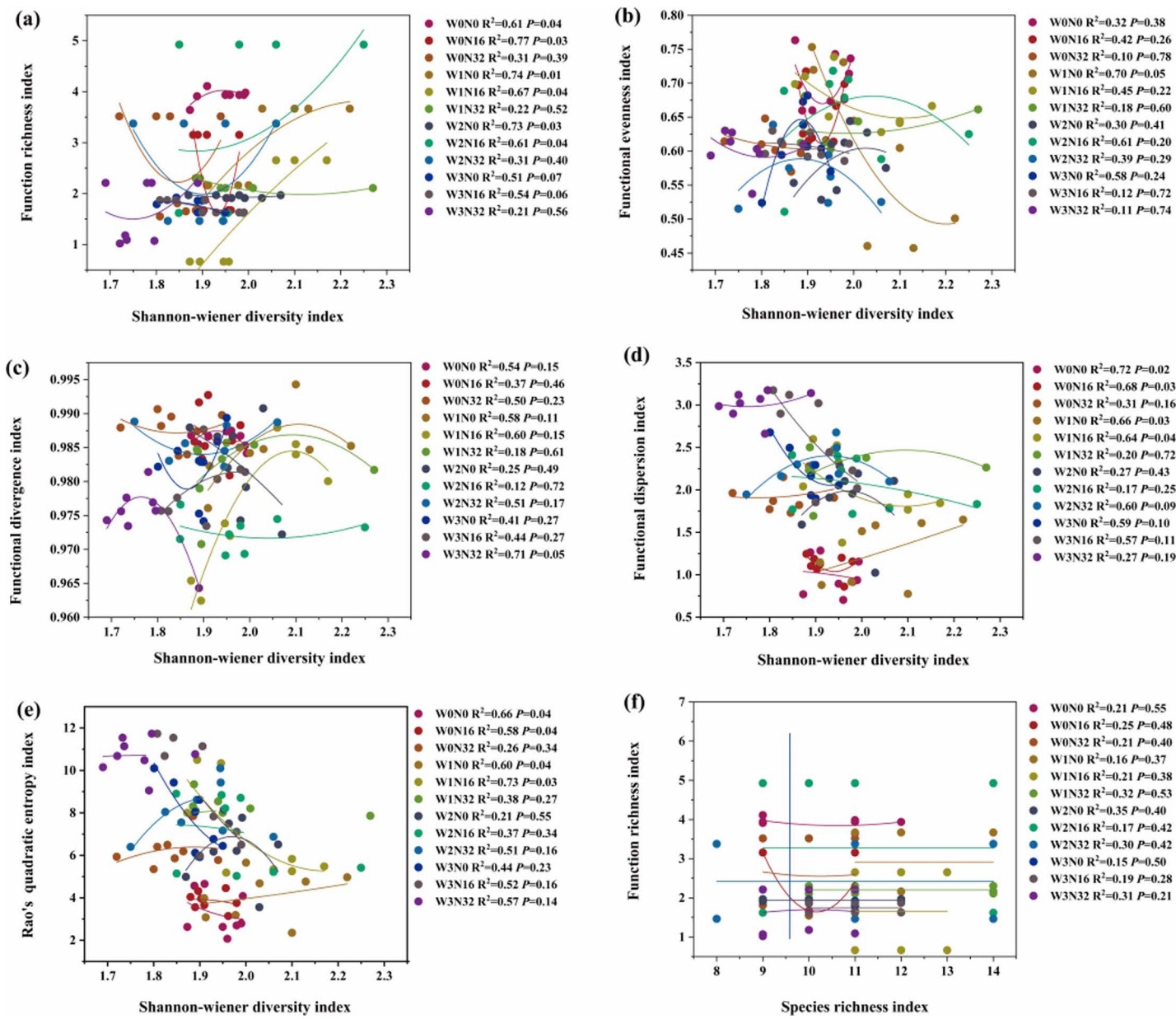

**Fig 4. Correlation between plant species diversity and plant functional diversity under warming and nitrogen deposition.**

warming, high-level warming has a greater impact on plant community species diversity. First of all, temperature is a key limiting factor for the growth of alpine plants, but excessive temperature may affect the growth of temperature-sensitive plant species [56]. Secondly, water availability is another key limiting factor for plant growth. Drying may lead to stomatal closure and photorespiration, thus affecting the growth of plant species [57,58]. Combined with the above, the difference in temperature and humidity in the process of warming will affect the plant community structure and species composition, but different plant species have different sensitivity to temperature. As a result, some plant species with poor resistance to temperature and stress exceeded the threshold of the adaptation range, resulting in a decrease in species dominance and loss of species, which eventually led to changes in community

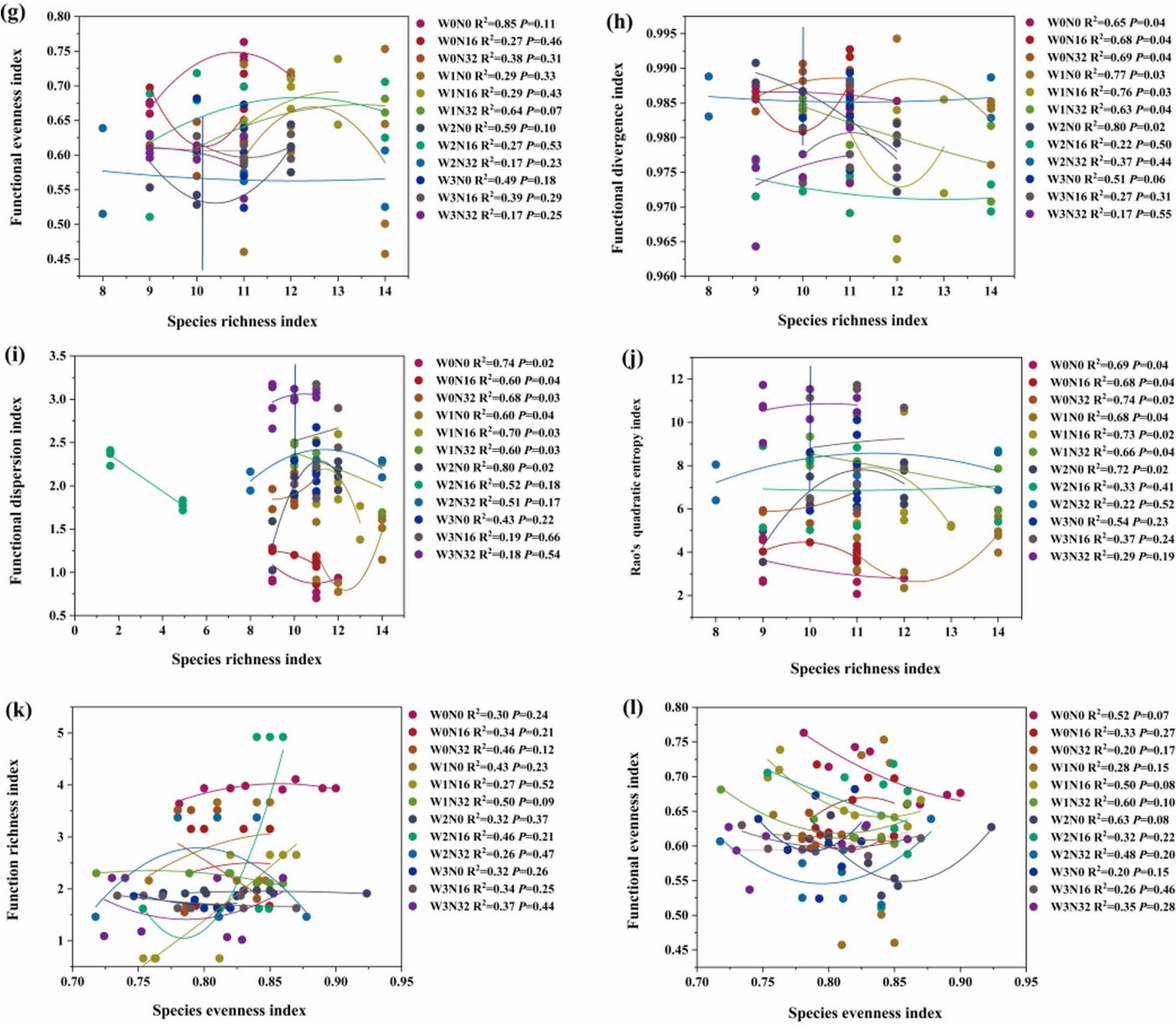

**Fig 4.** Continued.

structure [59]. The results of this study also showed that warming significantly increased the importance value of grasses and decreased the importance value of sedges and forbs. In the study of alpine meadow communities, it was also shown that warming increased the importance value of grasses and decreased the importance value of sedges [60]. Studies have also shown that plant communities will change from weed-based to grass-based under warming conditions, while sedge and legume species remain relatively unchanged in response to warming [61,62]. This is mainly because the response of different plant species to climate warming is mainly caused by the differences in biological characteristics and resource utilization of functional group plants and the response to environmental changes [63].

Many scholars believe that increased nitrogen deposition is one of the important factors affecting the species composition and diversity of plant communities [64,65]. This study

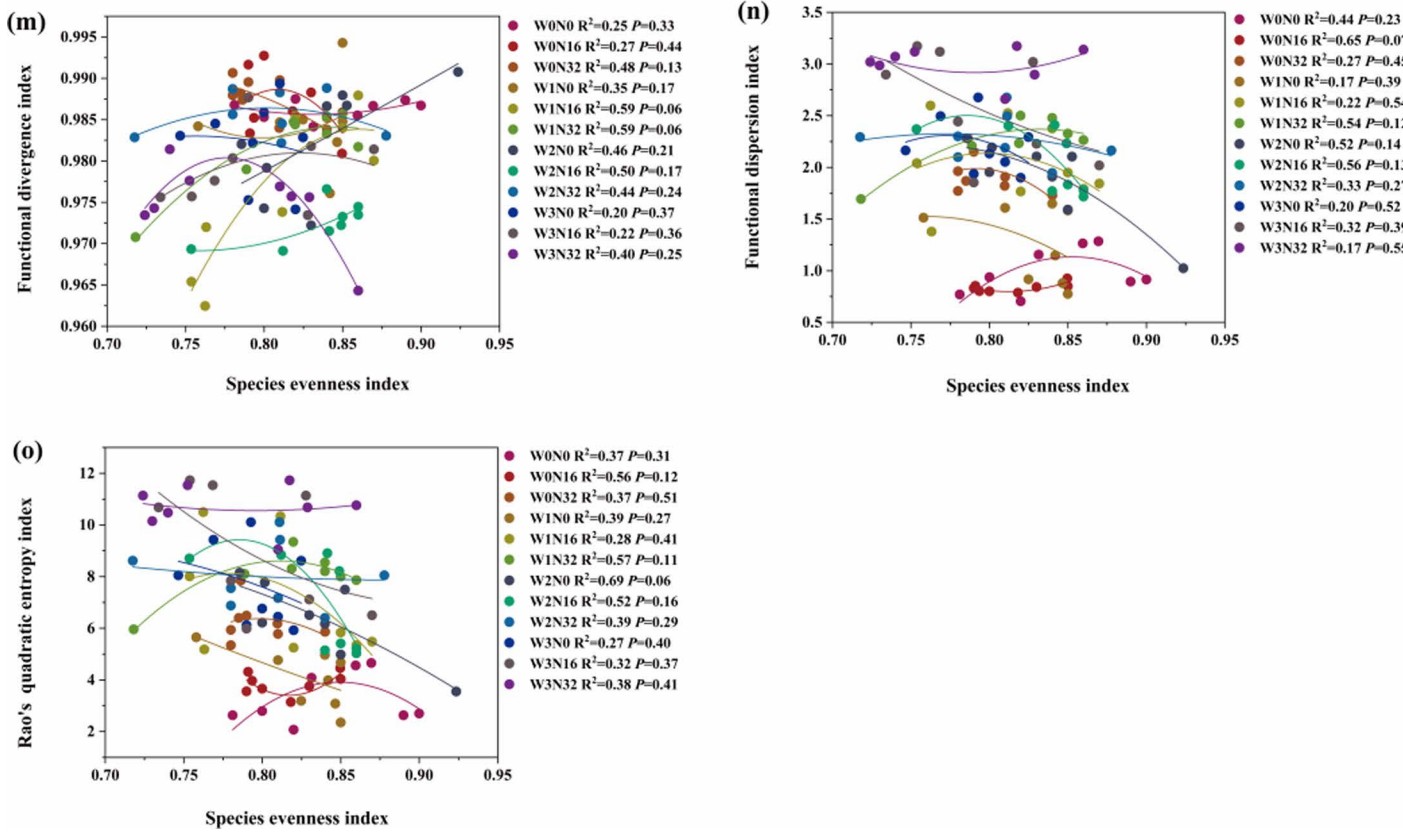

**Fig 4.** Continued.

found that in 2024 when the nitrogen addition amount was 32 kg. Nha$^{-1}$year$^{-1}$, the plant Shannon-Weiner index would be significantly reduced. Studies have shown that under a low nitrogen addition rate (≤10 g m$^{-2}$yr$^{-1}$), the negative effect of light competition on species richness is more significant, while under a high nitrogen addition rate (>10 g m$^{-2}$ yr$^{-1}$), the negative effect of soil ion toxicity is more significant, indicating that with the increase of soil nitrogen availability caused by nitrogen addition, the transition from light competition to ion toxicity occurs [66]. The results of this study showed that nitrogen deposition significantly increased the importance value of Grasses, but decreased the importance value of Sedges. Studies have shown that nitrogen deposition will enable dominant species to obtain more nutrients for rapid growth, thereby reducing the community species diversity index by reducing rare species [67]. The fierce competition within the plant community has prompted nitrogen-preferred plants to obtain more light and water resources. For example, higher species shade shorter species, and shorter plants are excluded from the community due to light restrictions, eventually leading to a decline in species diversity [8]. At the same time, the increase of soil available nitrogen may accelerate the growth of resource-acquired species, and the return on nutrient investment is faster, while inhibiting the growth of resource-saving species, resulting in a decrease in plant diversity [68].

The results of this study also showed that the interaction of warming and nitrogen deposition would significantly affect the Shannon-Wiener diversity index of plants, the importance value of forbs, and the importance value of grasses. Previous studies have also shown that plant species diversity is affected by the interaction of warming and nitrogen deposition [69]. The interaction between nitrogen deposition and warming is mainly attributed to the complex

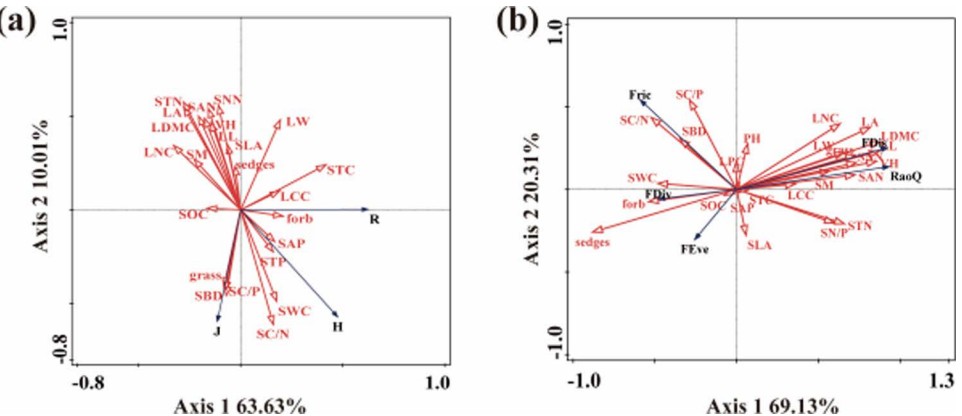

**Fig 5. Redundancy analysis of plant species diversity, plant functional diversity, plant functional traits and soil physical and chemical properties under warming and nitrogen deposition.**

and indirect effects of warming. On the one hand, due to the different sensitivity of species to temperature, the trait changes caused by warming are usually species-specific [70]. On the other hand, warming can affect soil nitrogen availability, which in turn affects plant growth [71].

## Effects of warming and nitrogen deposition on plant functional diversity

Plant functional traits are of great significance for exploring the relationship between plants and the environment, and provide a basis for predicting the response of ecosystems in the context of global change [72]. The results of this study showed that warming would increase the plant height-weighted mean, leaf area-weighted mean, leaf length-weighted mean, leaf-weight-weighted mean, and leaf nitrogen-weighted mean of the plant community. Climate warming will shift the mean of weighted traits of plant communities in a more accessible direction, including higher plants, larger leaves, and higher leaf nutrient concentrations [73,74]. This study area belongs to the permafrost region. Low temperature will hinder plant growth and produce conservative traits [75], while warming will reduce the temperature limit of plant species in alpine meadows and promote the change of plant leaf morphology (area, size, and width) [76].

The results of this study also showed that the weighted mean of leaf area, the weighted mean of leaf length, the weighted mean of leaf weight, and the weighted mean of leaf nitrogen increased significantly when the nitrogen addition amount was 32 kg. Nha-1year-1. Studies on the Qinghai-Tibet Plateau have also shown that nitrogen deposition can directly affect plant functional traits related to light acquisition or nitrogen availability (such as plant height, leaf area, plant carbon, nitrogen, phosphorus content, etc.) [77]. Nitrogen addition can increase the concentration of soil inorganic nitrogen, thereby increasing the reabsorption rate of plant nitrogen [78], which in turn increases the nitrogen content of plant leaves. To maintain the stoichiometric balance of plants, the demand for carbon and phosphorus increases with the increase in leaf nitrogen concentration [79]. At the same time, the plasticity of plant leaf functional traits enables plants to adapt to increased nitrogen deposition in the environment [80]. Previous studies have found that the increase of nitrogen deposition increases the nitrogen concentration of plant green and senescent leaf tissues, and also changes the stoichiometric ratio of plant leaves [81]. This study found that the interaction of warming and nitrogen deposition could significantly affect the functional traits of plant communities. Warming and

**Table 2. Redundancy analysis between plant diversity index, plant functional diversity index, soil physicochemical properties, and plant characteristics.**

| Factor | variable | Explain (%) | Contribution (%) | Pseudo-F | P |
|---|---|---|---|---|---|
| Plant species diversity index | STN | 6.8 | 16.8 | 6.8 | 0.008 |
| | CWM_LNC | 4.2 | 10.3 | 4.3 | 0.04 |
| | grasses | 17.7 | 8.0 | 7.2 | 0.001 |
| | SC/N | 3.7 | 9.2 | 4.3 | 0.044 |
| | CWM_LW | 2.4 | 6.0 | 2.9 | 0.053 |
| | SAN | 3.5 | 8.7 | 4.3 | 0.042 |
| | SWC | 1.6 | 4.0 | 2.0 | 0.064 |
| | forb | 1.3 | 3.1 | 1.6 | 0.231 |
| | CWM_VH | 1.3 | 3.2 | 1.6 | 0.214 |
| | CWM_SLA | 2.7 | 6.7 | 3.5 | 0.047 |
| | ST | 1.2 | 3.0 | 1.6 | 0.186 |
| | SBD | 0.6 | 1.5 | 0.8 | 0.404 |
| | STC | 0.6 | 1.4 | 0.7 | 0.406 |
| | SAP | 0.1 | 0.4 | 0.2 | 0.67 |
| | STEP | 0.3 | 0.8 | 0.4 | 0.54 |
| | SC/P | 0.8 | 2.1 | 1.1 | 0.332 |
| | SOC | 0.2 | 0.4 | 0.2 | 0.63 |
| | sedge | 0.1 | 0.3 | 0.1 | 0.716 |
| | CWM_LCC | <0.1 | 0.1 | <0.1 | 0.885 |
| | CWM_LL | 0.1 | 0.3 | 0.2 | 0.71 |
| | CWM_LA | 0.2 | 0.5 | 0.2 | 0.64 |
| | CWM_LDMC | <0.1 | 0.2 | 0.1 | 0.742 |
| | SNN | <0.1 | <0.1 | <0.1 | 0.882 |
| | CWM_LPC | <0.1 | <0.1 | <0.1 | 0.915 |
| | PH | <0.1 | <0.1 | <0.1 | 0.934 |
| Plant functional diversity index | grass | 54.2 | 66.0 | 110 | 0.002 |
| | SC/N | 10.5 | 12.7 | 27.3 | 0.009 |
| | CWM_LDMC | 3.6 | 4.4 | 10.3 | 0.012 |
| | SAN | 1.7 | 2.0 | 5.0 | 0.018 |
| | CWM_LA | 1.5 | 1.8 | 4.6 | 0.018 |
| | SWC | 1.4 | 1.7 | 4.5 | 0.017 |
| | SOC | 1.0 | 1.3 | 3.5 | 0.042 |
| | SC/P | 1.7 | 2.1 | 5.9 | 0.010 |
| | SN/P | 1.0 | 1.2 | 3.6 | 0.044 |
| | CWM_LCC | 0.8 | 1.0 | 2.9 | 0.061 |
| | CWM_LW | 0.7 | 0.9 | 2.7 | 0.078 |
| | ST | 0.5 | 0.6 | 1.9 | 0.214 |
| | STC | 0.4 | 0.4 | 1.4 | 0.278 |
| | CWM_LL | 0.4 | 0.5 | 1.4 | 0.228 |
| | PH | 0.3 | 0.4 | 1.2 | 0.284 |
| | CWM_LPC | 0.3 | 0.4 | 1.1 | 0.324 |
| | CWM_LNC | 0.3 | 0.3 | 1.0 | 0.327 |
| | forb | 0.3 | 0.4 | 1.2 | 0.288 |
| | CWM_VH | 0.2 | 0.2 | 0.7 | 0.482 |
| | CWM_SLA | 0.1 | 0.2 | 0.5 | 0.534 |

*(Continued)*

**Table 2.** (Continued)

| Factor | variable | Explain (%) | Contribution (%) | Pseudo-F | P |
|---|---|---|---|---|---|
| | SNN | 0.1 | 0.2 | 0.5 | 0.538 |
| | SAP | 0.2 | 0.2 | 0.6 | 0.498 |
| | STN | <0.1 | 0.1 | 0.4 | 0.571 |
| | SBD | <0.1 | <0.1 | 0.2 | 0.776 |
| | sedges | <0.1 | <0.1 | 0.2 | 0.764 |

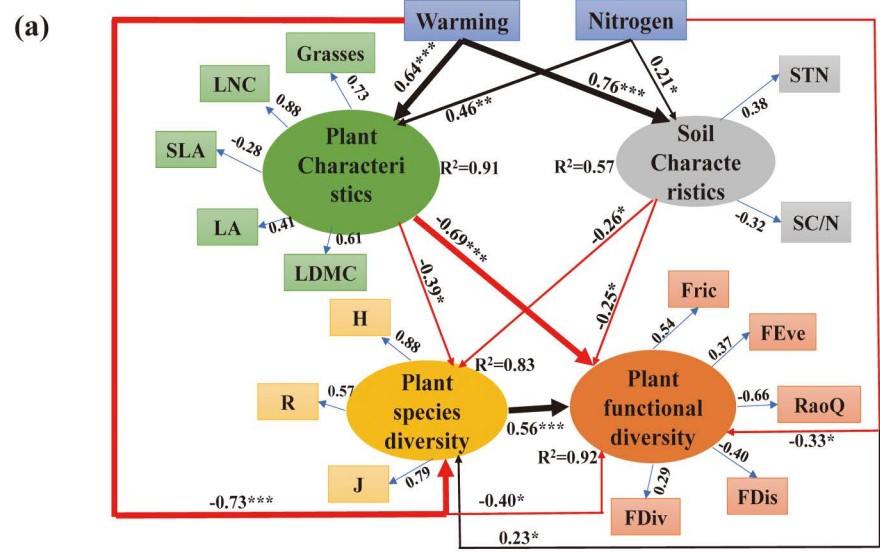

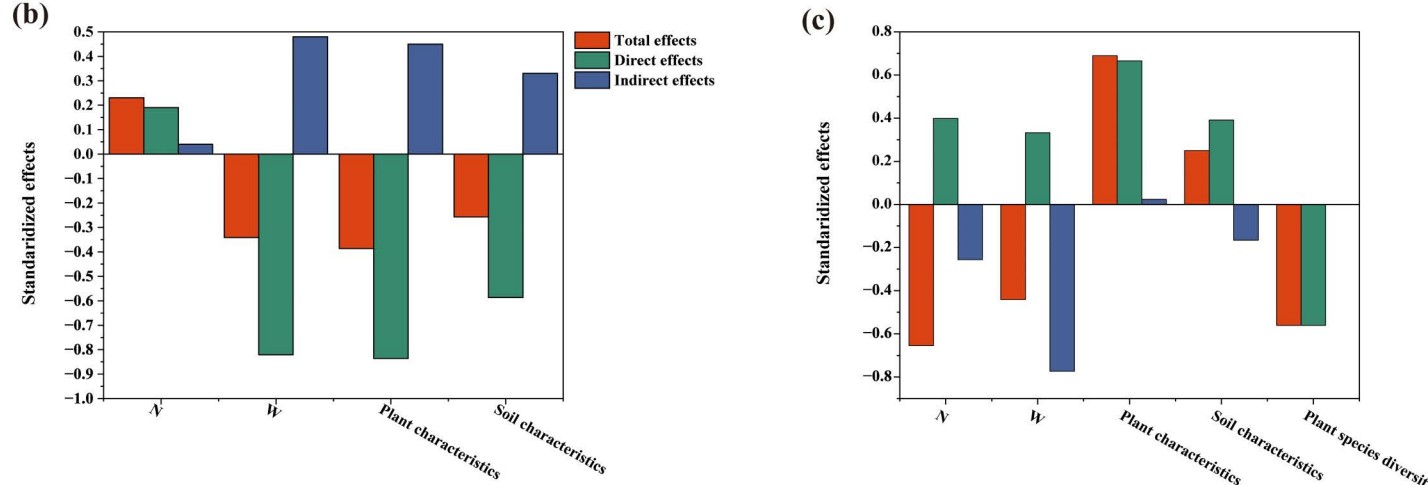

**Fig 6. The structural equation model showed the direct and indirect effects of plant factors and soil factors on plant species diversity and plant functional diversity. (a) Structural equation model; (b) The total effect, direct effect and indirect effect of plant and soil factors on plant species diversity. (c) The total effect, direct effect and indirect effect of plant and soil factors on plant functional diversity. Red and black arrows indicated a negative and positive relationship ($P < 0.05$). The width of the arrow is proportional to the strength of the path coefficient, and the adjacent numbers on the arrow represent the standardized path coefficient. R2 represents the proportion of variance explained by the model. \*\*\*, $P < 0.001$; \*\*, $P < 0.01$; \*, $P < 0.05$.**

nitrogen deposition have additive effects on plant community functional traits [82]. This is mainly due to the different sensitivity of plant species to temperature, and the trait changes caused by warming are usually species-specific [70]. At the same time, warming will increase the availability of soil nitrogen, thereby enhancing the effect of nitrogen enrichment, which indicates that the effects of nitrogen enrichment and warming on the functional traits of alpine meadows are superimposed [69].

Compared with the weighted mean of community traits, the functional diversity of plant communities indicates that there are differences in the strategies of plant acquisition and utilization of resources [83]. The results of this study showed that warming significantly reduced the plant functional richness index. Some studies have found that functional diversity is higher in stressful environments, while in resource-limited habitats, restricted similarity may tend to select species with more diverse resource utilization strategies [84]. In contrast, other studies have suggested that under malnourished conditions, functional diversity is low, because strong environmental filtering allows only certain species to persist, which may lead to a convergence of resource utilization strategies for co-existing species in the community [85]. This indicates that due to the need for plant species to adapt to external climate change, the functional traits of plant species are homogenized, which significantly reduces the niche of plant functional traits in the spatial range [86]. The RaoQ index and FDis index can be used to measure the degree of niche differentiation and resource competition among plants in the community [13]. The higher the index value, the stronger the niche complementarity between species, the weaker the competition, and the higher the resource utilization efficiency [87]. The results of this study showed that nitrogen deposition significantly reduced the plant functional richness index, Rao's quadratic entropy, and plant functional dispersion index. Biological processes such as competition can produce trait differentiation by excluding similar species (i.e., promoting niche differences) [69]. This means that the dominant plants tend to eliminate dwarf plants with similar traits in the vicinity. Nitrogen addition may lead to the emergence of new environmental conditions conducive to the selection of new traits while placing common species at a disadvantage and accelerating competitive exclusion [88,89].

## The relationship between plant species diversity and plant functional diversity under warming and nitrogen deposition

Plant species diversity and functional diversity are important foundations of ecosystem functions and important indicators for evaluating ecosystem stability [15]. The results of this study show that with the increase of warming gradient, the correlation between plant functional differentiation index, plant functional dispersion index, Rao's quadratic entropy, and species richness index is gradually weakened. Studies have also shown that plant species richness is closely related to functional richness [90]. Because when the temperature is not increased, the plant community has the highest species richness and the largest number of species. The wider the distribution of functional traits in the plant community, the greater the functional niche occupied by the species [91], and the correlation between the two is the strongest. However, with the increase in temperature, the number of species gradually decreased and the functional traits of species showed homogenization, which significantly reduced the spatial range of functional traits [92], and the correlation was weakened. The results of this study also showed that with the increase of warming gradient and nitrogen deposition, the correlation coefficient between plant Shannon-Wiener index and plant functional richness index, plant functional dispersion index, and Rao's quadratic entropy decreased gradually. Studies have shown that the relationship between plant species diversity and plant species functional diversity is different under different external environmental conditions [93]. With the increase of climate warming and nitrogen deposition, species that adapt to changes in the external

environment continue to grow, and these species have similar characteristics and show a high degree of redundancy in function. Studies have shown that grazing leads to the selection of palatable species, resulting in increased functional redundancy [49]. Functionally redundant species play the role of 'ecological insurance' and maintain the stability of the ecosystem [94]. In conclusion, as an important part of biodiversity, plant diversity plays an important role in maintaining the stability of grassland ecosystems. Phylogenetic diversity can reflect the ecological process of plant community composition and plays an important role in revealing the maintenance mechanism of biodiversity [95]. Therefore, exploring the effects of warming and nitrogen deposition on plant phylogenetic diversity in future studies can better understand the corresponding mechanisms of plant diversity on climate change in grassland ecosystems.

## Regulatory mechanisms of plant species diversity and plant functional diversity under warming and nitrogen deposition

The results of this study showed that the importance value of Grasses, the weighted mean of leaf nitrogen, the weighted mean of specific leaf area, the weighted mean of leaf area, and the weighted mean of leaf weight would have direct and indirect effects on plant species diversity and plant functional diversity through warming and nitrogen deposition. This may be because when temperature and nutrient limitations are alleviated, light resources often become an important factor in plant growth [96]. In this study area, the plant community under climate warming and nitrogen deposition is dominated by tall grasses and short herbs. Tall plants (*Poa pratensis* and *Elymus nutans*) usually have higher leaf area. A larger leaf area will increase the photosynthetic rate of plants, which gives tall plant species an advantage in light competition [80]. Dwarf and sheltered plants are limited by light, and the species diversity and functional diversity of plant communities change accordingly [97,98].

In addition, the results of this study also showed that soil total nitrogen was the main factor affecting plant species diversity and plant functional diversity. In the study of temperate grasslands, it was also found that soil total nitrogen and soil nitrogen-phosphorus ratio were the most important variables to explain the plant functional diversity index [31]. This is because nitrogen is the main limiting resource for plant productivity and the main factor affecting the growth of grassland plants [99]. Due to differences in nutrient utilization strategies between plant species or functional groups, changes in soil nitrogen content may change the competitive relationship between different species in the plant community and the balance of nutrient requirements, resulting in changes in plant community composition and structure [ 100,101]. When the soil is rich in nitrogen, plants can more easily absorb enough nitrogen, which promotes their growth and reproduction. This will lead to the rapid growth and expansion of some plant species in the plant community, forming a competitive advantage, that may suppress the growth of other plants and reduce plant species diversity [102,103]. On the contrary, if the soil contains low levels of nitrogen, plants will face nitrogen limitation, resulting in slower growth and less competition, which may provide more plant species with the opportunity to survive, thereby increasing species diversity [104].

The results of this study also showed that soil carbon nitrogen ratio was also an important factor affecting the species and functional diversity of plant communities. The study of 84 grasslands across six continents across a wide range of climatic gradients also found that plant diversity was positively correlated with soil carbon nitrogen ratio [105]. This is mainly due to the low nutritional value of organic matter with a high carbon-nitrogen ratio to microorganisms and slow decomposition [106]. Relatively scarce nitrogen resources may affect the competition between plants, thus affecting the diversity of plant communities. At the same time, warming and nitrogen deposition will affect the species and functional diversity of plant communities, and the diversity of plant species increases the cost of soil microbial decomposition

[107]. Very diverse organic compounds from different plant communities may decompose more slowly than less diverse organic matter from less diverse communities [108,109]. Soil factors determine which species or functional traits in plant communities will be retained, and this variability in plant species 'adaptation to the environment affects competition between communities and affects plant community structure [29]. Therefore, a better understanding of the relationship between plant community species diversity and soil physical and chemical properties will help to understand the overall function of grassland ecosystems better.

The results of this study also showed that with the increase of warming and nitrogen deposition, the plant species diversity index and plant functional diversity index decreased gradually, and there was a positive correlation between them. Some studies have also shown that the functional richness index is positively correlated with species richness [110,111], reflecting the occupation of niche space by existing species. The higher the richness, the more sufficient the niche space is occupied, the higher the community productivity, and the more stable the ecosystem function. This is mainly because warming and nitrogen deposition will reduce the niche dimension by alleviating resource constraints, which may aggravate species competition and lead to competitive exclusion, thereby reducing species diversity [112,113]. Recent studies have assessed changes in species richness from the perspective of functional traits and have shown that nitrogen-rich soils promote the growth of plant species with higher canopy and larger leaves, transforming underground nutrient competition into aboveground light competition [112,114,115]. The change of limiting factors is regarded as an indirect way to reduce the niche dimension, which leads to the reduction of functional diversity [116].

## Conclusion

In the study of alpine meadows on the Qinghai-Tibet Plateau, both warming and nitrogen deposition had significant negative effects on the plant species diversity index and plant functional diversity index. Plant factors (community importance value of grasses, community weighted mean of leaf nitrogen, community weighted mean of specific leaf area, and community weighted mean leaf weight) and soil environmental factors (soil total nitrogen and soil carbon-nitrogen ratio) under warming and nitrogen deposition will directly or indirectly affect plant community diversity. At the same time, the relationship between plant species diversity and functional diversity in the alpine meadow of the Qinghai-Tibet Plateau gradually weakened with the increase of warming and nitrogen deposition. Combined with the above, the changes in plant species and functional diversity under climate warming and nitrogen deposition in the future may reduce the stability of the alpine meadow ecosystem, which is challenging for alpine meadow grassland restoration and ecosystem management. Therefore, regular monitoring of changes in plant species diversity and functional diversity will help to detect changes in alpine meadow ecosystems promptly and take measures to address biodiversity loss. At the same time, monitoring soil health status, especially soil carbon-nitrogen ratio, nitrogen content, and other indicators, is of great significance for taking corresponding land management measures.

## Supporting information

**S1 Table. The results of the linear mixed effect model showed the effects of warming (W), nitrogen deposition (N), and their interaction (W × N) on plant species diversity in the first year (2023) and the second year (2024) of the experimental treatment.** The p-value ($< 0.05$) was significant. df denotes the degree of freedom.
(DOCX)

S2 Table. **The results of the linear mixed effect model showed the effects of warming (W), nitrogen deposition (N), and their interaction (W × N) on plant functional diversity in the first year (2023) and the second year (2024) of the experimental treatment.** The p-value (< 0.05) was significant. df denotes the degree of freedom.
(DOCX)

S3 Table. **The results of the linear mixed effect model showed the effects of warming (W), nitrogen deposition (N), and their interaction (W × N) on the weighted average of plant community functional traits in the first year (2023) and the second year (2024) of the experimental treatment.** The p-value (< 0.05) was significant. df denotes the degree of freedom.
(DOCX)

S1 Data. **Physical and chemical properties of plants and soils in various places.**
(XLSX)

## Author contributions

**Conceptualization:** Xuemei Xiang, Kejia De, Weishan Lin, Tingxu Feng, Fei Li, Xijie Wei.

**Data curation:** Xuemei Xiang, Kejia De, Weishan Lin, Tingxu Feng, Fei Li, Xijie Wei.

**Formal analysis:** Xuemei Xiang.

**Funding acquisition:** Xuemei Xiang.

**Investigation:** Xuemei Xiang, Tingxu Feng, Fei Li, Xijie Wei.

**Methodology:** Xuemei Xiang.

**Project administration:** Xuemei Xiang.

**Resources:** Xuemei Xiang.

**Software:** Xuemei Xiang.

**Supervision:** Xuemei Xiang, Xijie Wei.

**Validation:** Xuemei Xiang.

**Visualization:** Xuemei Xiang.

**Writing – original draft:** Xuemei Xiang, Kejia De.

**Writing – review & editing:** Xuemei Xiang, Kejia De.

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
