## [Decision Letter · Decision Letter 0]

22 Dec 2024

PONE-D-24-38030Soil Nutrients Explain Plant Community Species Diversity Changes Better Than Functional Traits Under Short-Term Warming and Nitrogen Deposition in Alpine MeadowsPLOS ONE

Dear Dr. DE,

Thank you for submitting your manuscript to PLOS ONE. After careful consideration, we feel that it has merit but does not fully meet PLOS ONE’s publication criteria as it currently stands. Therefore, we invite you to submit a revised version of the manuscript that addresses the points raised during the review process. In particular, I ask the authors to particularly consider the comments concerning the clarity of the methods and to be cautious with generalizations where the experiment is limited to one experimental site. Please submit your revised manuscript by Feb 05 2025 11:59PM. If you will need more time than this to complete your revisions, please reply to this message or contact the journal office at plosone@plos.org . Please include the following items when submitting your revised manuscript:

We look forward to receiving your revised manuscript.

Kind regards,

Francesco Boscutti

Academic Editor

PLOS ONE

Journal Requirements:

https://www.sciencedirect.com/science/article/abs/pii/S1574954121000017?via%3Dihub

In your revision ensure you cite all your sources (including your own works), and quote or rephrase any duplicated text outside the methods section. Further consideration is dependent on these concerns being addressed.

4. Please note that funding information should not appear in any section or other areas of your manuscript. We will only publish funding information present in the Funding Statement section of the online submission form. Please remove any funding-related text from the manuscript.

“This study was supported by the National Key Research and Development Project (2022YFD1602302), the Key R&D and Transformation Plan of Qinghai Provincial Science and Technology Department (2024-NK-137), and the Sanjiangyuan Ecosystem Field Observation and Research Station of the Ministry of Education (K9922050).”

6. PLOS requires an ORCID iD for the corresponding author in Editorial Manager on papers submitted after December 6th, 2016. Please ensure that you have an ORCID iD and that it is validated in Editorial Manager. To do this, go to ‘Update my Information’ (in the upper left-hand corner of the main menu), and click on the Fetch/Validate link next to the ORCID field. This will take you to the ORCID site and allow you to create a new iD or authenticate a pre-existing iD in Editorial Manager.

7. Please ensure that you include a title page within your main document. You should list all authors and all affiliations as per our author instructions and clearly indicate the corresponding author.

8. Please remove your figures from within your manuscript file, leaving only the individual TIFF/EPS image files, uploaded separately. These will be automatically included in the reviewers’ PDF.

Reviewers' comments:

Reviewer's Responses to Questions

**Comments to the Author**

1. Is the manuscript technically sound, and do the data support the conclusions?

Reviewer #1: Yes

Reviewer #2: Yes

Reviewer #3: Yes

2. Has the statistical analysis been performed appropriately and rigorously?

Reviewer #1: Yes

Reviewer #2: Yes

Reviewer #3: Yes

3. Have the authors made all data underlying the findings in their manuscript fully available?

Reviewer #1: Yes

Reviewer #2: Yes

Reviewer #3: Yes

4. Is the manuscript presented in an intelligible fashion and written in standard English?

Reviewer #1: Yes

Reviewer #2: Yes

Reviewer #3: Yes

5. Review Comments to the Author

Reviewer #1: This study explores the effects of short-term warming and nitrogen deposition on alpine meadow plant communities. Different warming and nitrogen addition treatments were set up at the Chengduo substation in Qinghai, and investigations and analyses were conducted on plant community structure, functional traits, and soil physicochemical properties. The results show that warming, nitrogen deposition, and their interaction have significant impacts on many aspects of the plant community, such as some functional traits, soil physicochemical indicators, and certain aspects of species diversity. The study found that soil nutrients can explain the changes in plant community diversity better than functional traits. Factors such as total soil nitrogen can explain its changes under warming and nitrogen deposition, providing references for related research and grassland management. Previous studies mainly discussed the responses of plant diversity in alpine grasslands on the Qinghai-Tibet Plateau to climate warming and nitrogen addition from the perspectives of environmental temperature and humidity, soil physicochemical properties, etc., while there are fewer studies from the perspective of plant functional traits （e.g., doi: DOI: 10.1007/s00344-016-9595-0; doi:10.1016/j.scitotenv.2023.168878）. Therefore, this study has certain innovation and is very worthy of publication. Nevertheless, before formal publication, the author is advised to consider the following aspects.

1. This study only explored the impact of warming and nitrogen addition on plant species diversity, but did not discuss the changes in plant phylogenetic diversity. Some previous studies have shown that external disturbances such as warming and nitrogen addition have different effects on plant species and phylogenetic diversity (e.g., doi:10.1016/j.ecoleng.2021.106331; doi:10.1016/j.ecolind.2021.108355; doi:10.1016/j.gecco.2023.e02464; doi:10.3389/fevo.2023.1126651; doi:10.3390/plants12173017; doi:10.1016/j.scitotenv.2023.168878). Therefore, it is recommended that the author add relevant discussions in the discussion section.

2. In addition to conventional soil physicochemical properties, soil microbial community diversity is closely related to plant diversity (e.g., doi:10.1016/j.scitotenv.2023.168878; DOI: 10.1016/j.gecco.2022.e02306). However, this study did not involve soil microorganisms. Moreover, in addition to plant functional traits, phyllosphere microorganisms of plants also have a relationship with plant diversity (e.g., doi:10.3390/plants12162891; doi:10.1016/j.jia.2024.06.008), but this study did not involve phyllosphere microorganisms either. Therefore, it is recommended that the author add discussions on the regulation of plant diversity by soil microbial diversity and phyllosphere microbial diversity.

Reviewer #2: The manuscript presents an experimental field study conducted in an alpine meadow in Qinghai-Tibet Plateau. In this study, researchers experimentally simulated the different levels of warming and nitrogen deposition and studied their effect on changes in plant community composition. They focused on the influence of functional traits and soil physicochemical properties on species richness, diversity and evenness under altering warming and nitrogen deposition. The study is interesting although it is conducted only on one alpine meadow, which limits authors to make some more general conclusions. It could be also interesting to observe the effects for longer time and evaluate temporal changes. I understand that multisite and temporal studies are not always possible to do, but it should be at least stated in discussion as study limitations. And this should be also reflected in the title of the paper. You cannot writhe “in Alpine meadows”, when your study was conducted only on one unique site. The topic of the study seems to be nowadays quite popular, but also very often studied and thus not very novel. Nevertheless, it is quite useful to get such information from different places and I also like the comparison of the influence of traits and soil properties. I think, it is quite interesting point of view.

In general, the topic is interesting, and manuscript is written well. Nevertheless, there are some things, I am not very happy with. Especially, the methods should be more appropriate – there are many missing details about experimental design, measurements and formulas. Some conceptual figure showing the experimental design could be helpful. In the whole manuscript, the names of variables and other terms are not unified and thus, it is very difficult to follow the text. There is a chaos in used formulas and the indexes and how you call them through the text. A part of the results is not presented in a well-arranged way. Some results are presented only in the text without any figure or table support, especially one factor results (in figures are results only for interactions, and values of statistical test with degrees of freedom and p values are missing almost everywhere). Arrangements of these values into a table could be helpful. There are more typing errors in the text, which I did not corrected.

Because the text does not have nor line numbers nor page numbers, it is very difficult to refer to something specific (and I have some suggestions). Thus, I am using at least the pdf page numbering, when necessary.

Abstract

“Additionally, Short-term nitrogen deposition affects plant community structure but has no significant impact on species diversity.” – It is not clear for me, what is the difference between community structure and diversity. It is not very clear nor from the further text. For me, community structure could be represented either by species richness or by diversity. It could also be the functional diversity or at least the proportion of functional groups. In any case, it is necessary to specify it more or write directly what do you mean.

Introduction

Page 10:

I do not like very much the first paragraph of the introduction, where you are listed everything together without going in detail (especially the part about functional group and nitrogen deposition). I prefer to describe it separately in individual paragraph with more details.

Page 11:

The paragraph about the nitrogen enrichment is introduced by response of diversity on climate change, but then you are writing only about the nitrogen deposition problematic. Here, the phrases from the previous paragraph about the effect of nitrogen deposition could be more suitable. And the concluding sentence of this paragraph is describing also warming, but it was not discussed here. May be, it could be useful to interconnect the effect of warming and nitrogen deposition. Or reversely, separate these two topics into two paragraphs.

What do you mean by “physiological traits”? Because in following text, you describe both morphological (SLA, leaf nitrogen concentration) and eco-physiological (photosynthetic rate) traits.

“Beyond leaf economics traits” – photosynthetic rate is not LES trait.

Page 12:

“Studies on the Qinghai-Tibet Plateau indicate that soil organic matter, available nitrogen, and phosphorus significantly affect plant diversity in alpine meadows [22].” – You are writing studies, but then you are citing only one study although it is not metanalysis. The same for more cases throughout the whole text (including discussion).

Materials and Methods

Page 14:

Elymus nutans and Poa annua are not forbs but grasses. And forbs (or grasses) are not family, but functional group. And is it Poa annua or P. pratensis. In Fig. 2, you have only P. pratensis.

50 x 50 m2 does not make sense. It is either 50 x 50 m or 2500 m2.

No warming (W0) treatment does mean that there was no enclosure?

Page 15:

I would like more details about soil and air temperature measurement. In which distance from the soil surface was the temperature measured (it is not enough to have it only in the caption of the figure)? How frequently the sensors measured the temperature? How many sensors did you used? One per one OTC or more?

What does “CK” mean?

Fig. 1:

a) and b) show changes in temperature for individual warming treatment. But before, you stated that you had the combination of plots with simulated warming with nitrogen deposition. Does it mean that these values are averages for W0N0, W0N16 and, W0N32, and so as? It should be stated in methodology.

Why there are axis labels in Chinese language?

The colour of columns of different treatment in c) and d) could be the same as line in a) and b). It would increase the readability of the figure.

For c) and d), it should be stated in the caption what do the letters above the columns mean? I expected that it is the result of Tukey test, but it should be stated in the caption of figure and also, this test should be described in methodology appropriately (what was compared with what).

Page 16:

How did you fit three 0.5 x 0.5 m quadrats into circle plot with diameter 0.95 m? Or I do not understand of the experimental design.

Page 17:

What does representative species for measurement of plant height mean? Do you mean five the most abundant species? How many individuals per species did you measure to get the plant height of one species?

What does the first formula before species richness index mean? It is not described anywhere else.

Page 18:

For trait measurement, you selected six from 15 recorded species. What does 15 recorded species mean? Fig. 2 shows that in total, there were 25 recorded species.

Why did you capitalize sedges, grasses and forbs?

You interchanged forbs and grasses. Poa pratensis is grass while the others are forbs.

Fig. 2:

The names of species should be in italic.

Page 19:

The formula for SLA is not correct and does not fit to the description. I think, the description is enough, and formula is not more necessary. On the other hand, there is not described how did you calculate LDMC.

What does subplot mean? Do you mean one 0.5 x 0.5 m quadrat? You did not write about it before as about subplot.

Is not the quadrat for soil sample the same quadrat as for vegetation and trait data? It is not clear for me.

Page 20:

What does importance value mean? It was not described anywhere else. How did you calculate it?

Page 21:

Do you used Tukey test only for interaction of warming and nitrogen deposition or also for these two factors individually? It is not clear from the methodology. From results, it seams that you tested both interaction and individual factors, but the results are stated only in figures only for interactions.

Page 22:

Vegan package should be cited.

Results

Describing the effect of warming and nitrogen deposition on CWM of traits, you did not present any statistical data (no results of statistical tests), only percentages and if it was or not significant. It should be somewhere, for example a table with the values for statistical test, df and p.

Page 23:

You cannot use once leaf dry weight and LDMC at another time. You should be consistent throughout the whole text.

Fig. 3, 4, 5:

You did not state what the values for W, N and W*N mean. What was the test and what are df?

You did not describe what letters mean. It is the result of Tukey but for what? And why did you not compare also the differences between only warming treatment? You describe it in text, but in figures, you have results only for interactions.

In Fig. 4, the y axis for pH (h) is interrupted, but it is not described nor in the caption of figure, nor in methodology.

Page 28:

The final sentence is more for discussion part than for results part.

Page 29:

There is no df for Monte Carlo permutation test (“(F = 2.2, P= 0.016)”).

Fig. 6:

c) Is it necessary to show so many decimals for in legend for Mantel´s r?

Discussion

The first parts of paragraphs contain very often quite introducing information which would fit more into introduction than to the discussion.

Page 32: I am confused about the community height and height measured as trait. It is not appropriately differed and from results, I had the impression that you did not use the community height. But now, it is in discussion, but I did not find nothing about community height in the results.

Page 34:

“Climate warming tends to reduce soil organic carbon and total nitrogen content [54].” This meta-analysis you cited showed that warming significantly increased soil dissolved organic C but did not change soil total organic C and soil total N. So, this does not support your statement.

Page 37:

“Our study indicates that the Shannon-Wiener index initially increases significantly with warming and then decreases…” – This formulation gives the impression that you measured warming in time and it is not true. You only have different levels of warming treatments.

“In the Loess Plateau, low-level warming increased plant community diversity, while high-level warming decreased it [67], suggesting that high-level warming has a more pronounced effect on species diversity [68].” – high-level warming had rather a more negative than more pronounced effect on species diversity.

You are stated that the water availability is a critical limiting factor for plant growth, but you are not discussing it with your results.

Supporting information

In table with data, there are some names of variables only in Chinese. It should be in English.

Reviewer #3: This article assesses the impact of warming and nitrogen deposition on plant traits, soil properties and species community assembly in an experiment carried out on alpine grasslands of the Qinghai-Tibet Plateau. I enjoyed reading this manuscript, the text is well written and the experiment represents a large amount of effort. The discussion section could be improved to link it better to the research questions. The graphs could also be clearer, particularly with reference to the results of the Tukey test. I have provided more details below and made some other suggestions for further improvements.

Materials and methods

In the section ‘Plant Community Structure Survey and Analysis’ the meanings of the letters C and B in the first formula are missing.

Some of the methods mentioned do not have a reference associated with them e.g. the molybdenum antimony anti-colorimetric method, hydrazine sulfate method and the indophenol blue method. Please add appropriate references for these techniques.

As well as increasing the temperature, open top chambers (OTCs) have been shown to alter the soil moisture and also protect plants against wind (see for example Hollister et al. 2023 https://doi.org/10.1139/as-2022-0030). Did you measure soil moisture in the experimental set up, before and after the placement of the OTCs? You mention in the discussion that reduced water availability also has an impact on community composition and growth so it could be this rather than (or in addition to) the warming that explains your results.

Results

Figure 3 – the figure legend does not match all the graphs presented. CWM_LW (leaf width), CWM_LPC (leaf phosphorus concentration) and CWM_LCC (leaf carbon concentration) are listed in the legend but are not part of the graphs. Instead CWM_LC/N and CWM_LC/P are in the graphs but not in the legend. Please correct.

Figures 3, 4 and 5 – it is not clear what the Tukey test letters refer to. Please add a note in the figure legend clarifying the difference between the upper and lower case letters.

Under the section “Effects of Warming and Nitrogen Deposition on Plant Community Species Diversity” the sentence “Nitrogen deposition also significantly affects the importance values of forbs and sedges” is repeated, please delete.

Discussion

At the beginning of the discussion it would be beneficial to have a short summary paragraph of the overall findings before going into the details. The discussion is presented in such a way that seems to highlight previous research findings before talking about your results. I would re-frame this to first emphasise the findings from your study and then explain the results in the context of previously published research.

At the end of the first paragraph of the discussion you mention “intraspecific trait responses to temperature changes vary among different species.”. Did you check for intraspecific variation in traits? It could be that there is a large trait variation within species or that one particular species is driving the changes observed.

Under warming and nitrogen deposition some traits did not change - the community-weighted means of leaf width, leaf phosphorus content, leaf carbon content, and the leaf carbon-to-phosphorus ratio. Do you have any speculation as to why this might be? It would be useful just to add a sentence in the discussion.

You mention that warming increases plant height. Do you think this will have a knock-on effect on herbivory e.g. the taller species will be more targeted by herbivores and so actually be at a disadvantage under warming? My understanding is that herbivory plays an important role in the Qinghai-Tibet Plateau but it is not mentioned in the manuscript. A paragraph about the potential impacts of your findings on herbivores would be interesting.

In the introduction you say “This study seeks to provide a theoretical basis for maintaining the species diversity and sustainability of plant communities in the alpine meadows of the Qinghai-Tibet Plateau under future climate warming and nitrogen deposition scenarios.”. However, I don’t see this mentioned in great detail in the discussion. It would be useful to have a paragraph about the implications of your results for future biodiversity in the Qinghai-Tibet Plateau and also any practical suggestions for how to maintain diversity in the future.

6. PLOS authors have the option to publish the peer review history of their article (what does this mean? ). If published, this will include your full peer review and any attached files.

**Do you want your identity to be public for this peer review?** For information about this choice, including consent withdrawal, please see our Privacy Policy .

Reviewer #1: No

Reviewer #2: No

Reviewer #3: No

---

## [Author Response · Author response to Decision Letter 1]

20 Jan 2025

Dear editors and reviewers:

Thank you for your letter and for the reviewers’ comments concerning our manuscript entitled “Soil Nutrients Explain Plant Community Species Diversity Changes Better Than Functional Traits Under Short-Term Warming and Nitrogen Deposition in Alpine Meadows” (Manuscript ID. PONE-D-24-38030). Those comments are all valuable and very helpful for revising and improving our paper, as well as the important guiding significance to our research. We have studied comments carefully and have made corrections which we hope meet with approval. The main corrections in the paper and the responses to the reviewers’ comments are as follows:

Reviewer1:

Question 1: This study only explored the impact of warming and nitrogen addition on plant species diversity, but did not discuss the changes in plant phylogenetic diversity. Some previous studies have shown that external disturbances such as warming and nitrogen addition have different effects on plant species and phylogenetic diversity (e.g., doi:10.1016/j.ecoleng.2021.106331; doi:10.1016/j.ecolind.2021.108355; doi:10.1016/j.gecco.2023.e02464; doi:10.3389/fevo.2023.1126651; doi:10.3390/plants12173017; doi:10.1016/j.scitotenv.2023.168878). Therefore, it is recommended that the author add relevant discussions in the discussion section.

Response: We recognize that phylogenetic diversity in plant communities is an important dimension of biodiversity. While this study focused primarily on species diversity and functional traits, we agree that phylogenetic diversity is also a key aspect that contributes to a deeper understanding of how warmth and nitrogen deposition affect plant communities. However, due to the diversity of development, this study did not directly measure system refer to the latest research (such as 10.1016 / j. coleng. 2021.106331), and has set up a file in the manuscript clearly points out the limitations, and emphasis on future research should consider this on the one hand, to provide a more comprehensive understanding of the responses of a community. Modified as follows (L634-L641) :

-In conclusion, as an important part of biodiversity, plant diversity plays an important role in maintaining the stability of grassland ecosystems. Phylogenetic diversity can reflect the ecological process of plant community composition and plays an important role in revealing the maintenance mechanism of biodiversity (Sun et al. 2021). Therefore, exploring the effects of warming and nitrogen deposition on plant phylogenetic diversity in future studies can better understand the corresponding mechanisms of plant diversity on climate change in grassland ecosystems.

Question 2: In addition to conventional soil physicochemical properties, soil microbial community diversity is closely related to plant diversity (e.g., doi:10.1016/j.scitotenv.2023.168878; DOI: 10.1016/j.gecco.2022.e02306). Therefore, it is recommended that the author add discussions on the regulation of plant diversity by soil microbial diversity

Response: We agree with reviewer #1 that soil microbial community diversity is closely related to plant diversity. While this study focused primarily on the effects of warmth and nitrogen additions on plant communities and soil physicochemical properties, we recognize that microbial communities play an important role in understanding plant diversity in alpine meadows. In response to this, we have added to the discussion section of the revised manuscript a discussion on the potential role of soil microbial community diversity in regulating plant diversity. We cite studies (DOI: DOI: 10.1016 / j.Geecco. 202.e02306) highlighting the effects of soil microorganisms on plant species diversity and functional status under climate change. We believe that these additions make the manuscript more complete. Thanks again to reviewer #1 for their valuable suggestions, which greatly enhanced the quality of the manuscript. Modified as follows (L676-L694) :

-The results of this study also showed that soil C/N ratio was also an important factor affecting the species and functional diversity of plant communities. The study of 84 grasslands across six continents across a wide range of climatic gradients also found that plant diversity was positively correlated with soil C/N ratio(Spohn et al. 2023). This is mainly due to the low nutritional value of organic matter with a high carbon-nitrogen ratio to microorganisms and slow decomposition(Cotrufo et al. 2013). Relatively scarce nitrogen resources may affect the competition between plants, thus affecting the diversity of plant communities. At the same time, warming and nitrogen deposition will affect the species and functional diversity of plant communities, and the diversity of plant species increases the cost of soil microbial decomposition(Han et al. 2022). Very diverse organic compounds from different plant communities may decompose more slowly than less diverse organic matter from less diverse communities (El Moujahid et al. 2017, Lehmann et al. 2020). Soil factors determine which species or functional traits in plant communities will be retained, and this variability in plant species ' adaptation to the environment affects competition between communities and affects plant community structure(Wang et al. 2022b). Therefore, a better understanding of the relationship between plant community species diversity and soil physical and chemical properties will help to understand the overall function of grassland ecosystems better.

We believe that these revisions have significantly improved the manuscript and have addressed all the issues raised. Thank you again for your valuable feedback.

Reviewer2:

Question 1: The manuscript presents an experimental field study conducted in an alpine meadow in Qinghai-Tibet Plateau. In this study, researchers experimentally simulated the different levels of warming and nitrogen deposition and studied their effect on changes in plant community composition. They focused on the influence of functional traits and soil physicochemical properties on species richness, diversity and evenness under altering warming and nitrogen deposition. The study is interesting although it is conducted only on one alpine meadow, which limits authors to make some more general conclusions. It could be also interesting to observe the effects for longer time and evaluate temporal changes. I understand that multisite and temporal studies are not always possible to do, but it should be at least stated in discussion as study limitations. And this should be also reflected in the title of the paper. You cannot writhe “in Alpine meadows”, when your study was conducted only on one unique site. The topic of the study seems to be nowadays quite popular, but also very often studied and thus not very novel. Nevertheless, it is quite useful to get such information from different places and I also like the comparison of the influence of traits and soil properties. I think, it is quite interesting point of view.

In general, the topic is interesting, and manuscript is written well. Nevertheless, there are some things, I am not very happy with. Especially, the methods should be more appropriate – there are many missing details about experimental design, measurements and formulas. Some conceptual figure showing the experimental design could be helpful. In the whole manuscript, the names of variables and other terms are not unified and thus, it is very difficult to follow the text. There is a chaos in used formulas and the indexes and how you call them through the text. A part of the results is not presented in a well-arranged way. Some results are presented only in the text without any figure or table support, especially one factor results (in figures are results only for interactions, and values of statistical test with degrees of freedom and p values are missing almost everywhere). Arrangements of these values into a table could be helpful. There are more typing errors in the text, which I did not corrected.

Because the text does not have nor line numbers nor page numbers, it is very difficult to refer to something specific (and I have some suggestions). Thus, I am using at least the pdf page numbering, when necessary.

Response: We greatly appreciate the reviewers' detailed and constructive feedback, which is of great value in improving the clarity and rigor of our manuscripts. The corresponding line numbers have been added in the revised draft. The following are our specific responses to the comments:

Question 2: Abstract“Additionally, Short-term nitrogen deposition affects plant community structure but has no significant impact on species diversity.” – It is not clear for me, what is the difference between community structure and diversity. It is not very clear nor from the further text. For me, community structure could be represented either by species richness or by diversity. It could also be the functional diversity or at least the proportion of functional groups. In any case, it is necessary to specify it more or write directly what do you mean.

Response: Thanks for your suggestion, the summary's mention that "short-term nitrogen deposition affects plant community structure but has no significant effect on species diversity" is really confusing. According to the suggestions of reviewers, the overall framework and results of the article are reorganized. Therefore, the summary part is modified. Modify as follows (L5-L24) :

-Plant species and functional diversity play an important role in the stability and sustainability of grassland ecosystems. However, the changes and mechanisms of plant species and functional diversity under warming and nitrogen deposition are still unclear. In this study, we investigated the plant and soil characteristics of alpine meadows on the Qinghai-Tibet Plateau to explore the changes in species and functional diversity of plant communities under warming and nitrogen deposition, as well as their interrelationships and key determinants. The results showed that warming, nitrogen deposition, and their interactions had significant effects on plant species diversity ( plant Shannon-Wiener index ) and functional diversity ( functional richness index, functional differentiation index, functional dispersion, and Rao's secondary entropy index ). With the increase of warming and nitrogen deposition, the Shannon-Wiener index of plants increased first and then decreased. The plant functional richness index, functional diversity index, functional dispersion index, and Rao's secondary entropy index showed a decreasing trend. At the same time, with the increase in temperature and nitrogen deposition, the relationship between plant species diversity index and functional diversity index in the alpine meadow of Qinghai-Tibet Plateau gradually weakened. Redundancy analysis and structural equation modeling showed that both warming and nitrogen deposition had significant negative effects on the plant species diversity index and plant functional diversity index. Plant factors ( Grasses importance value, leaf nitrogen weighted mean, specific leaf area-weighted mean, leaf area-weighted mean, and leaf weight weighted mean ) and soil environmental factors ( soil total nitrogen and soil carbon-nitrogen ratio ) directly or indirectly affect plant community diversity under warming and nitrogen deposition.

Question 3: Introduction

Page 10: I do not like very much the first paragraph of the introduction, where you are listed everything together without going in detail (especially the part about functional group and nitrogen deposition). I prefer to describe it separately in individual paragraph with more details.

Response: Thanks for your suggestion, rereading this paragraph does feel that the logic is not clear enough, and the impact of nitrogen deposition on plant diversity is only emphasized in this part. Therefore, read relevant literature, supplement relevant views, divide the paragraph into two paragraphs to make the structure clearer, and modify it as follows (L32-L62) :

-Global surface temperatures are projected to rise by approximately 4°C by 2100 (Wang et al. 2020). Simultaneously, anthropogenic nitrogen emissions and deposition are increasing rapidly on a worldwide scale(Kwaku et al. 2021). In the past few decades, the combined effects of climate change factors such as temperature rise and nitrogen deposition have led to the degradation of the function and structure of grassland ecosystems, resulting in accelerated desertification, loss of biodiversity, and changes in carbon balance(Chen et al. 2022). The loss of biodiversity not only means the extinction of species but also poses a threat to food security, social stability, and human survival through the food chain and food web.

Plant community diversity is an important part of biodiversity(Zha et al. 2022). Plant species diversity reflects the complex relationship between biology, environment, and biological resource richness(Katovai et al. 2012). Research has shown that global warming significantly reduces plant species richness and diversity in terrestrial ecosystems (Yang et al. 2011). Different plant functional groups respond differently to climate warming (Fay et al. 2011). The effects of nitrogen deposition on plant species diversity vary depending on the nitrogen use efficiency and adaptability of the plant community (Roth et al. 2015, Zhang et al. 2015, Kwaku et al. 2021). In the study of temperate grassland and semi-arid grassland, it was found that plant species richness showed a downward trend under nitrogen deposition (Zhang et al. 2014, He et al. 2016). With the increase in temperature and nitrogen deposition, the number of vascular plants is generally increasing, which may be because temperature and nutrients are the main limiting factors of alpine ecosystems(Boutin et al. 2017). Plant functional diversity takes into account the redundant and complementary functions within the plant community, as well as the response of functional traits to environmental stress or disturbance(Li et al. 2024). Species diversity and functional diversity are key determinants of grassland ecosystem stability (Petchey and Gaston 2006, Dunck et al. 2016). Therefore, exploring the feedback mechanism of plant community diversity on climate warming and nitrogen deposition is of great scientific value and practical guiding significance for scientifically understanding the maintenance mechanism of grassland ecosystem diversity on the Qinghai-Tibet Plateau under climate change and guiding grassland management.

Question 4: Page 11:

What do you mean by “physiological traits”? Because in following text, you describe both morphological (SLA, leaf nitrogen concentration) and eco-physiological (photosynthetic rate) traits.

“Beyond leaf economics traits” – photosynthetic rate is not LES trait.

Response: Thanks for your suggestion, after reading the relevant literature, it is indeed found that the photosynthetic rate is not an economic trait of leaves, and this part has been modified. Modify as follows (L73-L75) :

- Studies have shown that nitrogen enrichment enhances specific leaf area, and leaf nitrogen concentration in plant species (Eskelinen and Harrison 2015, La Pierre and Smith 2015).

Question 5: Page 12:

“Studies on the Qinghai-Tibet Plateau indicate that soil organic matter, available nitrogen, and phosphorus significantly affect plant diversity in alpine meadows [22].” – You are writing studies, but then you are citing only one study although it is not metanalysis. The same for more cases throughout the whole text (including discussion).

Response:Thank you for your suggestion. This paragraph mainly focuses on the discussion that the physical and chemical properties of soil have a significant impact on plant diversity, but the conclusions reached by different studies are inconsistent. Therefore, it is necessary to explore the effects of soil physicochemical properties on plant diversity under climate warming and nitrogen deposition. Read the relevant literature and rearrange the section. Modified as follows (L93-L106) :

-In temperate grassland, soil pH, total nitrogen, and nitrogen-phosphorus ratio are the key factors to explain plant functional diversity(Rodríguez et al. 2023). Studies have also shown that

---

## [Decision Letter · Decision Letter 1]

5 Feb 2025

Effects of warming and nitrogen deposition on species and functional diversity of plant communities in the alpine meadow of Qinghai-Tibet Plateau

PONE-D-24-38030R1

Dear Dr. DE,

We’re pleased to inform you that your manuscript has been judged scientifically suitable for publication and will be formally accepted for publication once it meets all outstanding technical requirements.

Kind regards,

Francesco Boscutti

Academic Editor

PLOS ONE

Additional Editor Comments (optional):

Reviewers' comments:

Reviewer's Responses to Questions

**Comments to the Author**

1. If the authors have adequately addressed your comments raised in a previous round of review and you feel that this manuscript is now acceptable for publication, you may indicate that here to bypass the “Comments to the Author” section, enter your conflict of interest statement in the “Confidential to Editor” section, and submit your "Accept" recommendation.

Reviewer #1: (No Response)

Reviewer #2: All comments have been addressed

Reviewer #3: All comments have been addressed

2. Is the manuscript technically sound, and do the data support the conclusions?

Reviewer #1: (No Response)

Reviewer #2: Yes

Reviewer #3: Yes

3. Has the statistical analysis been performed appropriately and rigorously?

Reviewer #1: (No Response)

Reviewer #2: Yes

Reviewer #3: Yes

4. Have the authors made all data underlying the findings in their manuscript fully available?

Reviewer #1: (No Response)

Reviewer #2: Yes

Reviewer #3: Yes

5. Is the manuscript presented in an intelligible fashion and written in standard English?

Reviewer #1: (No Response)

Reviewer #2: Yes

Reviewer #3: Yes

6. Review Comments to the Author

Reviewer #1: The author has made a large number of revisions and I have no further comments. It is recommended to accept it

Reviewer #2: A revised the manuscript entitled “Effects of warming and nitrogen deposition on species and functional diversity of plant communities in the alpine meadow of Qinghai-Tibet Plateau” is significantly improved compared to the previous version of the manuscript. The authors made a great job on it. I am satisfied with the answers on my questions and with incorporating my suggestions into the text of the manuscript.

I have only few minor suggestions:

While you defined species diversity indices, you did not do it for functional diversity indexes (L 251-253). I think it is not necessary to write the formulas but at least citations of the relevant papers should be there to know what you are referring to.

L 274 – you are firstly mentioned the structural equation models and in following text, you are using the abbreviation SEM. But this abbreviation was not stated before, so it should be there.

The Results part - it would be more comprehensive if you are referring to the Figures and Tables throughout the whole text and not only at the end of each paragraph. You also should refer not only to the whole Figures but also to the individual panels (a, b, c…) of Figures.

Fig.2 – the figure is two time there. In the caption, L 331, you stated “The same bellow”. What does it mean? If you mean that it is the same for next Figures, I think it is not the best solution. I prefer to repeat the proper caption with detail for each Figures (Fig. 3, 4, Table 2). Or another solution is to refer to these Figures in the captions (e.g., abbreviations same as in Fig.2).

L 516-517 – “mainly” used two times is not necessary.

L 555 – two times “leaf length-weighted mean”.

Reviewer #3: (No Response)

7. PLOS authors have the option to publish the peer review history of their article (what does this mean? ). If published, this will include your full peer review and any attached files.

**Do you want your identity to be public for this peer review?** For information about this choice, including consent withdrawal, please see our Privacy Policy .

Reviewer #1: No

Reviewer #2: No

Reviewer #3: No

---

## [Editor Report · Acceptance letter]

PONE-D-24-38030R1

PLOS ONE

Dear Dr. DE,

I'm pleased to inform you that your manuscript has been deemed suitable for publication in PLOS ONE. Congratulations! Your manuscript is now being handed over to our production team.

Kind regards,

on behalf of

Dr. Francesco Boscutti

Academic Editor

PLOS ONE